# Uev1A counteracts oncogenic *Ras* stimuli in both polyploid and diploid cells

Qi Zhang[1†], Yunfeng Wang[1†], Xueli Fu[2†], Ziguang Wang[1†], Yang Zhang[1], Lizhong Yan[1], Yuejia Wang[1], Muhan Yang[1], Dongze Song[1], Ruixing Zhang[2], Hongru Zhang[2,3]*, Shian Wu[1]*, Shaowei Zhao[1]*

[1]Department of Genetics and Cell Biology, College of Life Sciences, Nankai University, Tianjin, China; [2]Department of Biochemistry and Molecular Biology, College of Life Sciences, Nankai University, Tianjin, China; [3]Nankai International Advanced Research Institute (SHENZHEN FUTIAN), Shenzhen, China

## eLife Assessment

This **valuable** study examines the role of E2 ubiquitin enzyme, Uev1a in tissue resistance to oncogenic RasV12 in *Drosophila melanogaster* polyploid germline cells and human cancer cell lines. The **solid** evidence suggests that Uev1a works with the E3 ligase APC/C to degrade Cyclin A. This work would be of interest to researchers in germline biology and cancer.

**\*For correspondence:**
hrzhang@nankai.edu.cn (HZ);
wusa@nankai.edu.cn (SW);
swzhao@nankai.edu.cn (SZ)

[†]These authors contributed equally to this work

**Competing interest:** See page

**Abstract** Oncogenic *Ras* is known to induce DNA replication stress, leading to cellular senescence or death. In contrast, we found that it can also trigger polyploid *Drosophila* ovarian nurse cells to die by inducing aberrant division stress. To explore intrinsic protective mechanisms against this specific form of cellular stress, here, we conducted a genome-wide genetic screen and identified the E2 enzyme Uev1A as a key protector. Reducing its expression levels exacerbates the nurse cell death induced by oncogenic *Ras*, while overexpressing it or its human homologs, UBE2V1 and UBE2V2, mitigates this effect. Although Uev1A is primarily known for its non-proteolytic functions, our studies demonstrate that it collaborates with the E3 APC/C complex to mediate the proteasomal degradation of Cyclin A, a key cyclin that drives cell division. Furthermore, Uev1A and UBE2V1/2 also counteract oncogenic *Ras*-driven tumorigenesis in diploid cells, suppressing the overgrowth of germline tumors in *Drosophila* and human colorectal tumor xenografts in nude mice, respectively. Remarkably, elevated expression levels of UBE2V1/2 correlate with improved survival rates in human colorectal cancer patients harboring oncogenic *KRAS* mutations, indicating that their upregulation could represent a promising therapeutic strategy.

## Introduction

Although the human body contains trillions of cells that could potentially be targeted by oncogenic mutations, cancers arise infrequently throughout a human lifetime, suggesting the presence of effective resistance mechanisms (*Lowe et al., 2004*). One key mechanism involves the induction of DNA replication stress by these oncogenic mutations, leading to cellular senescence or death (*Hills and Diffley, 2014*; *Kotsantis et al., 2018*). This mechanism is mediated through the activation of the DNA damage response (DDR) pathway and the tumor suppressor protein p53, which are triggered by DNA double-strand breaks (DSBs). Only the cells that escape senescence and death, such as those harboring *p53* mutations, can undergo transformation by oncogenic mutations, thereby initiating tumorigenesis (*Gaillard et al., 2015*; *Igarashi et al., 2024*; *Macheret and Halazonetis, 2015*). Among the most frequently mutated oncogenes in human cancers are the *RAS* genes (*KRAS*, *NRAS*,

and *HRAS*), which mutations are present in 20–30% of all cancer cases (*Gimple and Wang, 2019*; *Sanchez-Vega et al., 2018*). These mutations often occur at codons 12, 13, and 61, resulting in RAS small GTPases being locked in a constitutively active, GTP-bound state (*Moore et al., 2020*). Notably, previous studies have shown that oncogenic $HRAS^{G12V}$ can induce DNA replication stress via the DDR pathway, ultimately leading to cellular senescence (*Di Micco et al., 2006*; *Serrano et al., 1997*).

The *Ras oncogene at 85D* gene (*Ras85D*, hereafter referred to as *Ras*) in *Drosophila* exhibits high homology to human *RAS* genes (*Neuman-Silberberg et al., 1984*). Our previous research demonstrated that oncogenic $Ras^{G12V}$ triggers cell death in *Drosophila* ovarian nurse cells (*Zhang et al., 2024b*), a type of post-mitotic germ cells that undergo G/S endoreplication to become polyploid (*Hammond and Laird, 1985*; *Figure 1A*). Notably, oncogenic $Ras^{G12V}$ promotes the division of mitotic germ cells, the precursors to nurse cells. Furthermore, monoallelic deletion of the *cyclin A* (*cycA*) or *cyclin-dependent kinase 1* (*cdk1*) gene ($cycA^{+/-}$ or $cdk1^{+/-}$) suppresses $Ras^{G12V}$-induced nurse cell death (*Zhang et al., 2024b*). While CycA, a key cyclin promoting cell division, is not expressed in normal nurse cells (*Lilly et al., 2000*; *Lilly and Spradling, 1996*), its ectopic expression in nurse cells can trigger their death (as observed in this study). These findings suggest that the nurse cell death induced by oncogenic $Ras^{G12V}$ is primarily due to aberrant promotion of cell division. *Drosophila* ovarian nurse cells thus offer a valuable model for studying this specific form of cellular stress.

Ubiquitination not only mediates protein degradation through the proteasome or lysosome but also plays crucial regulatory roles in various biological processes (*Dikic, 2017*; *Kwon and Ciecha-nover, 2017*). These functions are largely determined by the topological structures of the polyubiquitin chains, where ubiquitin can be linked to another ubiquitin via any of its seven lysine (K) residues or its first methionine residue. Among these, K11- and K48-linked polyubiquitin primarily mediates the proteasomal degradation. In contrast, K63 linkage facilitates the lysosomal degradation of protein aggregates and damaged organelles through autophagy. Additionally, K63 linkage is involved in non-degradative processes such as DNA repair, kinase activation, and protein transport (*Kwon and Ciecha-nover, 2017*). Ubc13 is a ubiquitin-conjugating (E2) enzyme that specifically catalyzes the formation of the K63 linkage. However, this activity requires a unique cofactor known as Ubc variant (Uev). Uev proteins lack the catalytic cysteine residue necessary for ubiquitin thioester formation; instead, they work with Ubc13 to form a functional E2 complex (*Hofmann and Pickart, 1999*; *McKenna et al., 2001*). These E2 enzymes have been shown to play regulatory roles in various biological processes, including cell death (*Ma et al., 2013*), DNA repair (*Broomfield et al., 1998*), DDR, and innate immunity (*Andersen et al., 2005*; *Bai et al., 2020*; *Zhou et al., 2005*). However, it still remains unknown whether they also possess proteasomal proteolytic functions and whether such functions hold significant biological roles.

In this study, to investigate the intrinsic protective mechanisms against $Ras^{G12V}$-induced nurse cell death in *Drosophila*, we conducted a genome-wide genetic screen and identified Uev1A as a crucial protector. Mechanistically, Uev1A works in conjunction with the anaphase-promoting complex or cyclosome (APC/C) to degrade the essential cyclin CycA via the proteasome, highlighting the critical role of its proteolytic function. Additionally, Uev1A and its human homologs, UBE2V1 and UBE2V2, also counteract oncogenic *Ras*-driven tumorigenesis in diploid cells, inhibiting the overgrowth of *Drosophila* germline tumors and human colorectal tumor xenografts in nude mice, respectively. These findings suggest that upregulation of UBE2V1/2 could represent a promising therapeutic approach for human colorectal cancers with *RAS* mutations.

## Results

### Genetic screen identifies Uev1A as a crucial protector against $Ras^{G12V}$-induced nurse cell death

While oncogenic $Ras^{G12V}$ triggers cell death in nurse cells, it does not have the same effect in cystocytes, the precursor cells before differentiating into nurse cells (*Zhang et al., 2024b*). To confirm $Ras^{G12V}$ expression in cystocytes, we generated a *UASz-flag-Ras^{G12V}* transgenic fly strain, which phenocopied the effects observed with the previously used *UASp-Ras^{G12V}*. RAS small GTPases need to be anchored to the inner cell membrane for their proper function (*Willingham et al., 1980*). Using *bam-GAL4-VP16* (*Chen and McKearin, 2003*), we overexpressed $Ras^{G12V}$ specifically in cystocytes (*bam>flag-Ras^{G12V}*) at 29°C and observed membrane-localized Flag signals, validating the normal expression of $Ras^{G12V}$

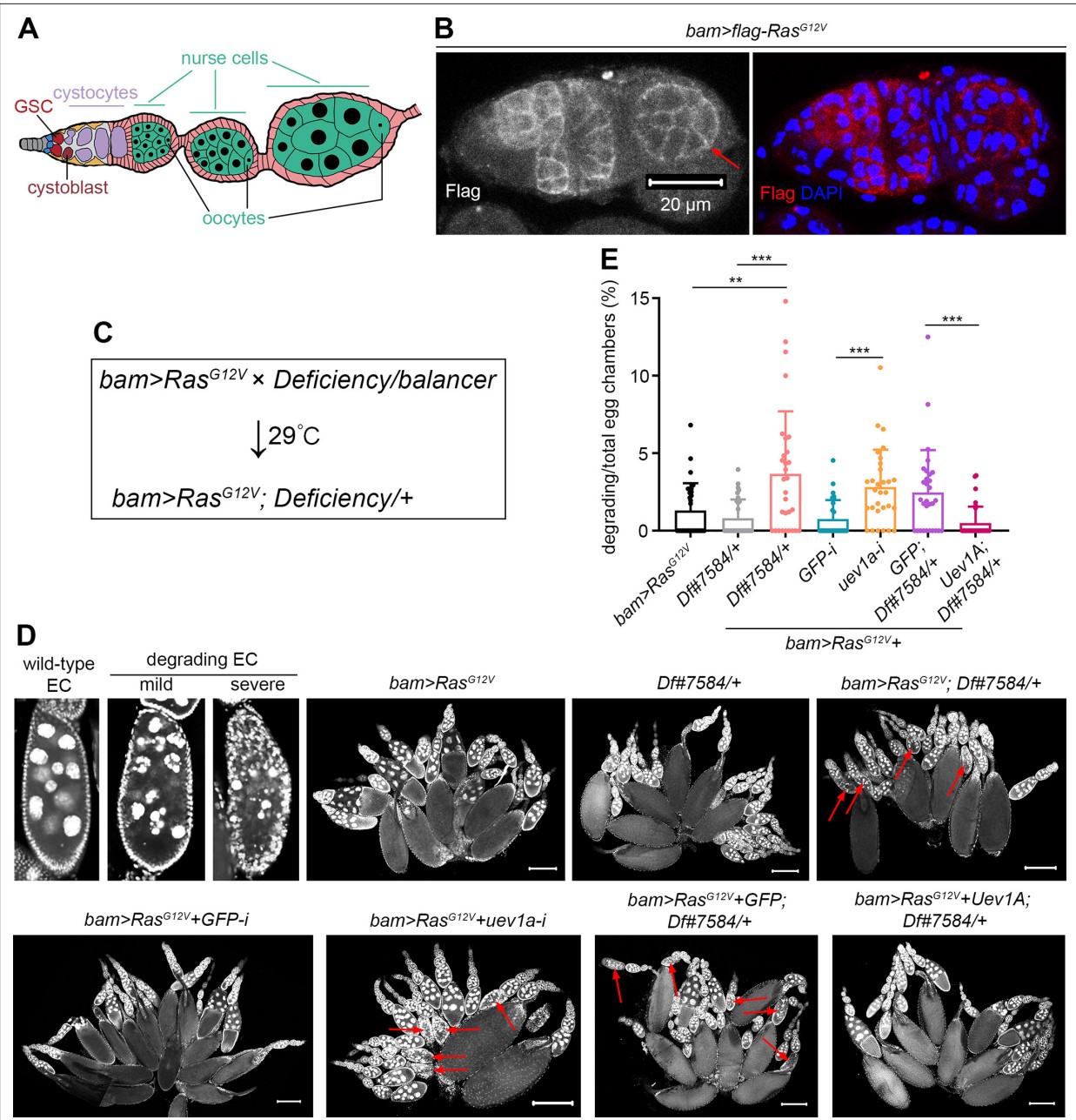

**Figure 1.** Genetic screen identifies Uev1A as a crucial protector against *Ras*<sup>G12V</sup>-induced nurse cell death. (**A**) Schematic cartoon for *Drosophila* ovariole. During oogenesis, germline stem cell (GSC) undergoes asymmetric division to generate two daughter cells: one that self-renews and maintains GSC identity, and the other, called a cystoblast, that differentiates to support oogenesis. As differentiation progresses, each cystoblast performs four rounds of division with incomplete cytokinesis to produce 16 interconnected cystocytes, establishing a germline cyst. This germline cyst is then surrounded by epithelial follicle cells to form an egg chamber. Within each egg chamber, one of the 16 germ cells becomes the oocyte, while the remaining 15 differentiate into nurse cells. These nurse cells undergo G/S endocycling, becoming polyploid to aid in oocyte development. (**B**) Representative germarium and early-stage egg chamber with Flag-Ras*<sup>G12V</sup>* overexpression driven by *bam-GAL4-VP16*. The red arrow denotes an early-stage egg chamber. (**C**) Genetic screening strategy. Genotype of '*bam>RasG*<sup>G12V</sup>': *bam-GAL4-VP16/FM7;; UASp-Ras*<sup>G12V</sup>*/TM6B*. (**D**) Representative ovaries and egg chambers (DAPI staining). The red arrows in (**D**) denote degrading egg chambers. Scale bars: 200 µm. (**E**) Quantification data. 30 ovaries from 7-day-old flies were quantified for each genotype. Statistical significance was determined using t test (groups = 2) or one-way ANOVA (groups >2): ** (p<0.01) and *** (p<0.001).

(*Figure 1B*). Also, Flag signals were detected in some early-stage nurse cells, although these signals gradually diminished over time (*Figure 1B*). During cell death, nurse cell nuclei progress through distinct morphological stages: from large and round, to disorganized and condensed, and finally to completely fragmented into small, spherical structures (*Figure 1D*). Given the interconnection and synchronous death of all nurse cells within an egg chamber, we quantified this death phenotype at the egg-chamber level. Notably, nurse cell death remained very low in *bam>flag-Ras$^{G12V}$* fly ovaries (*Figure 1D and E*). This may be attributed to either insufficient levels of the oncoproteins or the presence of a protective mechanism.

To explore the potential protective mechanism, we conducted a genetic modifier screen by introducing individual genome-wide *Deficiencies* (361 lines) into *bam>RasG$^{G12V}$* flies (*bam>RasG$^{G12V}$*; *Deficiency/+*, *Figure 1C*). Of note, ovaries from *bam>RasG$^{G12V}$*; *Df#7584/++* flies exhibited the highest incidence of degrading egg chambers with dying nurse cells per ovary (see *Source data 1*). One gene deleted in this deficiency line is *uev1a*, and RNAi targeting *uev1a* reproduced the phenotype seen with *Df#7584* (*Figure 1D and E*). In this and all subsequent experiments, we quantified the nurse-cell-death phenotype using the percentage of degrading to total egg chambers per ovary (*Figure 1E*), a method that is more precise than our approach in the genetic screen. To further verify the role of *uev1a*, we generated a *UASz-uev1a* transgenic fly strain. Overexpression of Uev1A, but not GFP (*UASp-GFP*; *Zhang et al., 2024a*), successfully rescued the nurse-cell-death phenotype in *bam>RasG$^{G12V}$*; *Df#7584/++* fly ovaries (*Figure 1D and E*), confirming that Uev1A is a crucial protector against *Ras$^{G12V}$*-induced nurse cell death.

## Protective role of Uev1A against the nurse cell death induced by direct overexpression of Ras$^{G12V}$

In contrast to the scenario above (*bam>RasG$^{G12V}$*), direct overexpression of Ras$^{G12V}$ in nurse cells using *nos-GAL4-VP16* (*Rørth, 1998*; *Van Doren et al., 1998*) (*nos>RasG$^{G12V}$*) resulted in substantial cell death even at 25°C (*Figure 2A and D*; *Zhang et al., 2024b*). Such observation prompted us to investigate the role of Uev1A in this context. Notably, the incidence of dying nurse cells was markedly elevated in nos>RasG$^{G12V}$+uev1a-RNAi ovaries compared to the nos>RasG$^{G12V}$+GFP-RNAi control ovaries (*Figure 2A and D*). To further validate this, we generated two uev1a mutants using the CRISPR/Cas9 technique: uev1a$^{Δ1}$ and uev1a$^{Δ2}$. Both mutations consist of small deletions that cause frame shifts within the second coding exon of the uev1a gene (*Figure 2B*). Of note, flies with the uev1a$^{Δ1/Δ1}$, uev1a$^{Δ1/Δ2}$, uev1a$^{Δ1}$/Df#7584, uev1a$^{Δ2/Δ2}$, or uev1a$^{Δ2}$/Df#7584 genotype could not survive into adults, suggesting that both mutations are strong loss-of-function alleles. Consistent with the effects observed with uev1a-RNAi, either mutation markedly increased the incidence of dying nurse cells in nos>RasG$^{G12V}$ ovaries (*Figure 2C and D*).

We next investigated whether upregulating Uev1A could mitigate the nurse cell death induced by direct overexpression of *Ras$^{G12V}$*. Remarkably, compared to the *nos>RasG$^{G12V}$+lacZ* control ovaries, the incidence of dying nurse cells was significantly reduced in *nos>RasG$^{G12V}$+Uev1A* ovaries (*Figure 2F and G*). Uev1A has two human homologs, UBE2V1 and UBE2V2, which share 67% (85%) and 67% (86%) sequence identities (similarities) with Uev1A, respectively (*Figure 2E*). We generated transgenic fly strains for both *UASz-UBE2V1* and *UASz-UBE2V2*. Notably, overexpression of either UBE2V1 or UBE2V2, but not lacZ (*UASz-lacZ*; *Zhang et al., 2024b*), significantly mitigated *Ras$^{G12V}$*-induced nurse cell death, similar to the effects observed with Uev1A overexpression (*Figure 2F and G*). Taken together, these results further confirm the protective role of Uev1A against *Ras$^{G12V}$*-induced nurse cell death.

Then, we were intrigued by the potential of Uev1A to protect against the nurse cell death induced by other oncogenic mutations. Yorkie (Yki), a key oncoprotein in the Hippo pathway (*Huang et al., 2005*), has a hyperactive form known as Yki$^{3SA}$ (*Oh and Irvine, 2009*). Its overexpression (*nos>Yki$^{3SA}$*; *UASz-Yki$^{3SA}$*; *Zhang et al., 2024a*) at 29°C, but not at 25°C, could induce nurse cell death (*Figure 2—figure supplement 1*), albeit much less severe than that induced by *nos>RasG$^{G12V}$* at 25°C (compared with *Figure 2A and D*). Of note, the nurse cell death induced by oncogenic *Yki$^{3SA}$* was significantly alleviated by Uev1A overexpression (*Figure 2—figure supplement 1*), implying a broad role of Uev1A in this process. However, due to the mild effect of Yki$^{3SA}$ on triggering nurse cell death, we focused on Ras$^{G12V}$ in our subsequent studies.

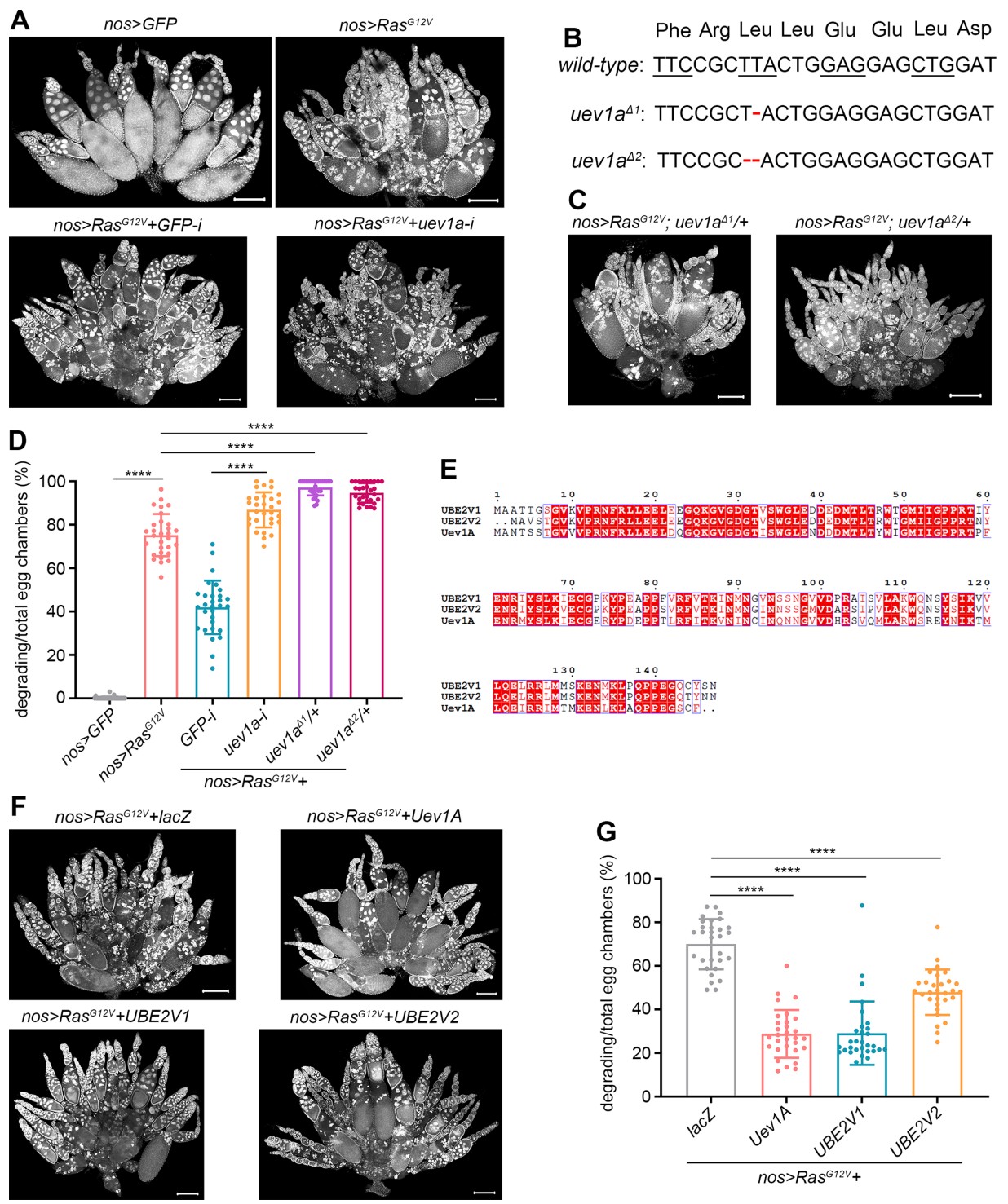

**Figure 2.** Uev1A protects against the nurse cell death induced by direct overexpression of Ras$^{G12V}$. (**A, C, and F**) Representative samples (DAPI staining). (**B**) Molecular information of the *uev1a$^{Δ1}$* and *uev1a$^{Δ2}$* mutations. The red dashed lines represent nucleotide deletions. (**D and G**) Quantification data. 30 ovaries from 3-day-old flies were quantified for each genotype. Statistical significance was determined using t test (groups = 2) or one-way ANOVA (groups>2): **** (p<0.0001). (**E**) Protein sequence alignment of Uev1A, UBE2V1, and UBE2V2. It was performed using CLUSTALW and ESPript 3.0 software.

The online version of this article includes the following figure supplement(s) for figure 2:

**Figure supplement 1.** Uev1A protects against *Yki$^{3SA}$*-induced nurse cell death.

## The DDR pathway and p53 play opposite roles in *Ras*^G12V-induced nurse cell death

To investigate how Uev1A protects against *Ras*^G12V-induced nurse cell death, we first sought to further explore the mechanisms driving this cell death. Since egg chambers contain both nurse cells and oocytes (*Figure 1A*), we considered the possibility that the death signal could originate from oocytes. The *Bicaudal D* (*BicD*) gene is essential for oocyte determination, and egg chambers deficient in it fail to specify oocytes (*Suter and Steward, 1991*). We confirmed this by using *nos>BicD*-RNAi (*Figure 3—figure supplement 1A*). Importantly, nurse cell death was much more pronounced in *nos>BicD-RNAi+RasG*^G12V egg chambers than in *nos >BicD-RNAi+GFP* control ones (*Figure 3— figure supplement 1A and B*), indicating that the death signal is intrinsic to nurse cells rather than originating from oocytes.

Oncogenic RAS can induce DNA replication stress in diploid cells, thereby activating the DDR pathway and p53 to trigger cellular senescence or death (*Di Micco et al., 2006*; *Serrano et al., 1997*).

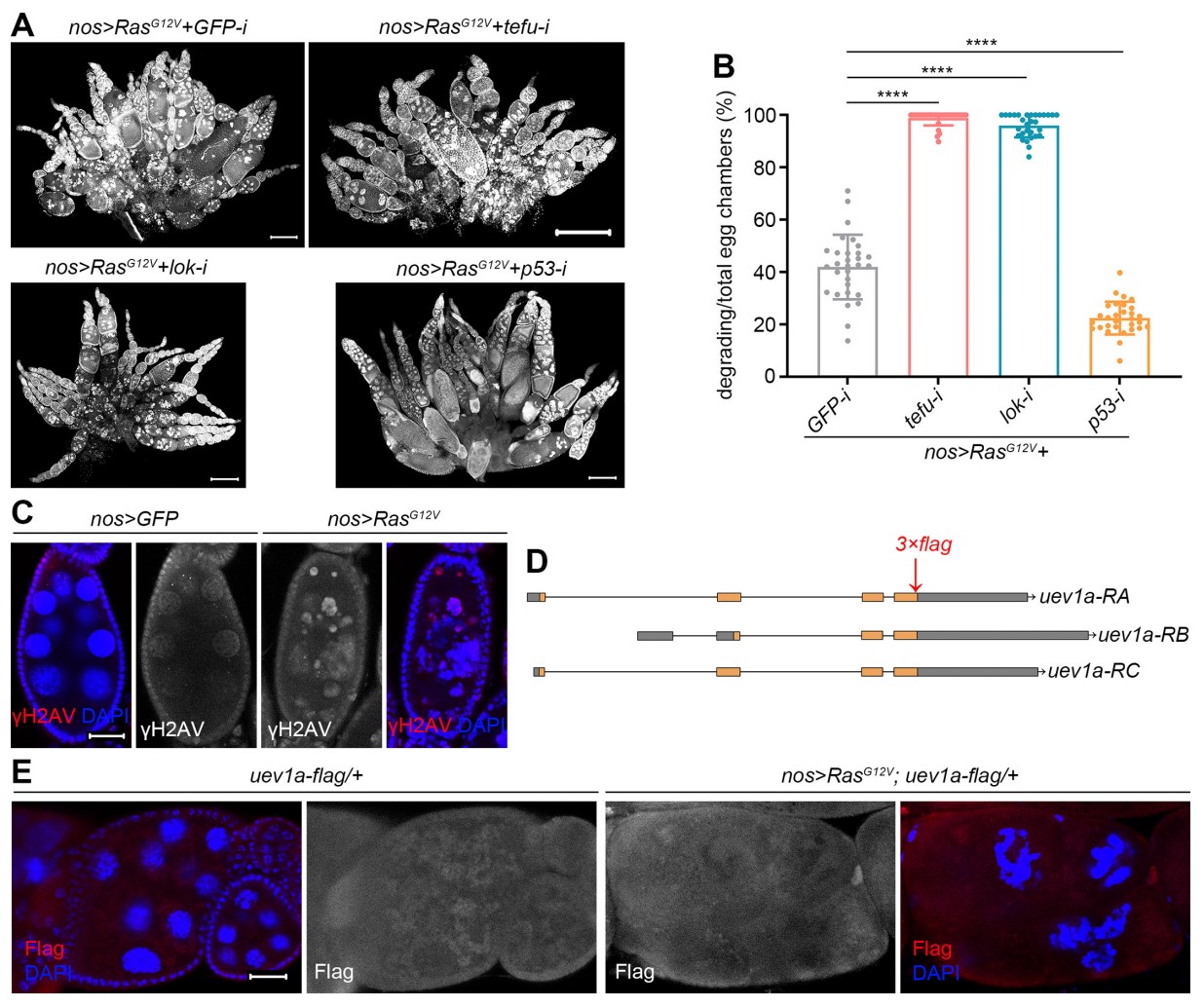

**Figure 3.** Roles of the DNA damage response (DDR) pathway and p53 in *Ras*^G12V-induced nurse cell death. (**A**) Representative ovaries (DAPI staining). Scale bars: 200 μm. (**B**) Quantification data. 30 ovaries from 3-day-old flies were quantified for each genotype. Statistical significance was determined using one-way ANOVA: **** ($p<0.0001$). (**C and E**) Representative samples. Scale bars: 20 μm. (**D**) Schematic cartoon for *uev1a-flag* knock-in.

The online version of this article includes the following figure supplement(s) for figure 3:

**Figure supplement 1.** Oncogenic *Ras*^G12V intrinsically triggers nurse cell death.

**Figure supplement 2.** Uev1A is expressed in stretch follicle cells.

**Figure supplement 3.** Uev1A does not directly degrade the Ras^G12V oncoproteins.

To investigate whether DDR plays a similar role in $Ras^{G12V}$-induced nurse cell death, we targeted two key DDR genes: *telomere fusion* (*tefu*, encoding *Drosophila* ATM) (*Oikemus et al., 2004*; *Silva et al., 2004*; *Song et al., 2004*) and *loki* (*lok*, encoding *Drosophila* Chk2) (*Masrouha et al., 2003*; *Xu et al., 2001*). Strikingly, knockdown of either gene markedly enhanced this cell death (*Figure 3A and B*), revealing a protective role for the DDR pathway. By contrast, knockdown of *p53* suppressed this cell death (*Figure 3A and B*), demonstrating opposing functions for DDR and p53 in this context. To assess DNA DSBs and the ensuing DDR, we monitored the phosphorylation of γH2AV, the *Drosophila* histone variant analogous to mammalian H2AX (*Lake et al., 2013*). As egg chamber developmental stages are difficult to discern during degradation, we compared size-matched egg chambers, which are typically stage-matched under normal conditions and have comparable antibody penetration. Elevated γH2AV staining was observed in degrading *nos>RasG^{G12V}* egg chambers compared to *nos>GFP* controls (*Figure 3C*), indicating a heightened burden of DNA DSBs.

Given that RNAi targeting *tefu* or *lok* phenocopied that of *uev1a*, we hypothesized that Uev1A protects against $Ras^{G12V}$-induced nurse cell death through the DDR pathway. If so, Uev1A may also be upregulated in response to $Ras^{G12V}$-driven stress. To test this, we initially attempted to generate an antibody against Uev1A using its full protein sequence as the antigen; however, this approach was unsuccessful. As an alternative, we created a *uev1a-flag* knock-in fly strain by inserting the coding sequence of a 3xFlag tag immediately before the stop codon ('TAG') of the *uev1a* gene (*Figure 3D*). These knock-in flies were homozygous viable and fertile, indicating that the Flag insertion did not disrupt Uev1A's normal function. Previous studies have shown that Uev1A can activate JNK signaling (*Ma et al., 2013*), which acts in the surrounding follicle cells to promote nurse cell removal during late oogenesis (*Timmons et al., 2016*). Indeed, we detected Uev1A-Flag signals in such follicle cells (*Figure 3—figure supplement 2*), confirming that these signals reflect endogenous Uev1a expression in *Drosophila* ovaries. Surprisingly, Uev1A-Flag signals were low in both *nos>RasG^{G12V}*; *uev1a-flag/++* and *uev1a-flag/++* control nurse cells, with no significant differences between the two groups (*Figure 3E*). This suggests that Uev1A expression remains at basal levels, rather than being upregulated by $Ras^{G12V}$-driven stress in this context. Although we cannot rule out the possibility that Uev1A may play a role in the DDR pathway at these basal levels, these findings prompted us to explore its potential proteolytic function as an E2 enzyme.

Following the Occam's Razor principle, we first investigated whether Uev1A is involved in downregulating $Ras^{G12V}$ oncoprotein levels. To address the challenges of analyzing membrane-localized signals in *nos>flag-Ras^{G12V}* nurse cells due to severe cell death, we performed this assay in *bam>flag-Ras^{G12V}* ovaries at 29°C. Notably, similar membrane-localized Flag signals were detected in both *bam>flag-Ras^{G12V}+Uev1A* and *bam>flag-Ras^{G12V}+GFP* control germ cells, including nurse cells (*Figure 3—figure supplement 3*). This result suggests that Uev1A does not downregulate $Ras^{G12V}$ oncoprotein levels.

## Uev1A downregulates CycA protein levels in $Ras^{G12V}$-induced dying nurse cells

Our previous research demonstrated that oncogenic $Ras^{G12V}$ promotes germline stem cell (GSC) over-proliferation by activating the mitogen-activated protein kinase (MAPK) pathway (*Zhang et al., 2024b*). This finding led us to investigate whether the same pathway influences $Ras^{G12V}$-induced nurse cell death. Notably, knockdown of *downstream of raf1* (*dsor1*, encoding *Drosophila* MAPKK) (*Tsuda et al., 1993*) or *rolled* (*rl*, encoding *Drosophila* MAPK) (*Biggs and Zipursky, 1992*) significantly mitigated $Ras^{G12V}$-induced nurse cell death (*Figure 4A and B*), highlighting the pivotal role of the MAPK pathway in this process.

The Ras/MAPK pathway is well known to promote cell cycle progression (*Gimple and Wang, 2019*). Our previous work demonstrated that reducing the gene dosage of *CycA* or *Cdk1* suppresses $Ras^{G12V}$-induced nurse cell death (*Zhang et al., 2024b*). In addition to CycA and Cdk1, Cyclin B (CycB) and String (Stg) are two other essential regulators that drive cell division (*Figure 4C*; *Edgar and Lehner, 1996*). To determine if individual overexpression of these regulators could induce nurse cell death, we tested each and found that only CycA triggered this phenotype (*Figure 4C and D*). Notably, this cell death was suppressed by co-overexpression of CycA and Uev1A (*Figure 4C and D*), indicating a genetic interaction between them. In line with previous findings (*Lilly et al., 2000*; *Lilly and Spradling, 1996*), CycA protein was undetectable in wild-type endocycling nurse cells when assessed using an anti-CycA antibody (*Whitfield et al., 1990*; *Figure 4E*). In stark contrast, it was abundantly

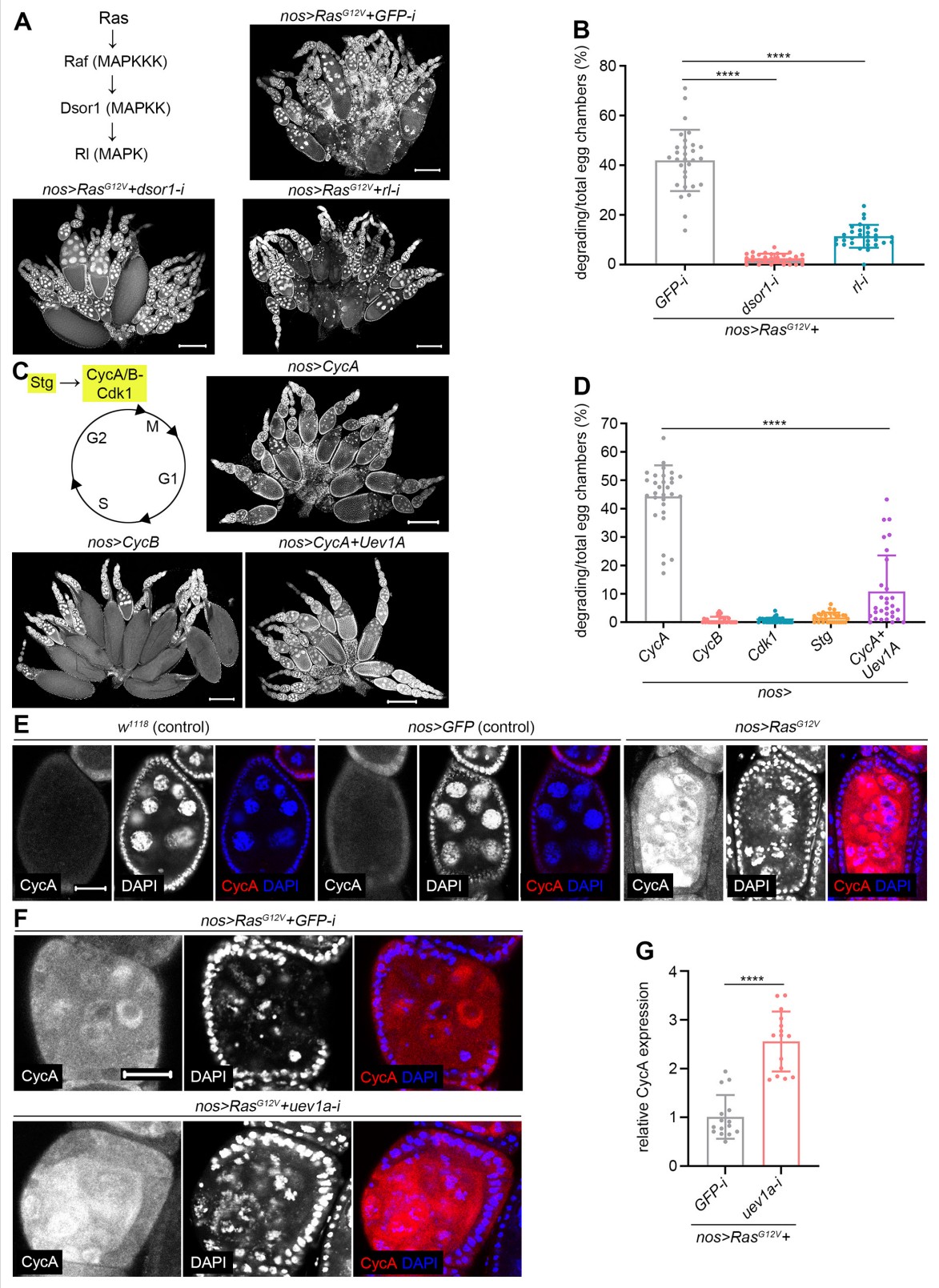

**Figure 4.** Uev1A collaborates with CycA to mitigate *Ras^{G12V}*-induced nurse cell death. (**A, C, E, and F**) Representative ovaries. DAPI staining in (**A and C**). Scale bars: 200 μm in (**A and C**), 20 μm in (**E and F**). (**B, D, and G**) Quantification data. 30 ovaries (**B and D**) and 15 size-matched egg chambers (**G**) from 3-day-old flies were quantified for each genotype. Statistical significance was determined using t test (groups = 2) or one-way ANOVA (groups >2): **** (p<0.0001).

expressed in dying $nos>RasG^{G12V}$ nurse cells (**Figure 4E**), underscoring a critical role for CycA in this cell death process. Additionally, we assessed CycA protein levels in size-matched $nos>RasG^{G12V}$ nurse cells under either *uev1a* or *GFP* (control) knockdown condition. Notably, *uev1a* knockdown increased CycA levels compared with the controls (**Figure 4F and G**), demonstrating that Uev1A downregulates CycA protein levels in $Ras^{G12V}$-induced dying nurse cells.

## Uev1A collaborates with the APC/C complex to mitigate $Ras^{G12V}$-induced nurse cell death

It is well established that the APC/C complex primarily functions as an E3 ligase to facilitate the degradation of CycA during cell cycle progression (**Sudakin et al., 1995**). Thus, a compelling model to explain our findings is that Uev1A collaborates with the APC/C complex to degrade CycA.

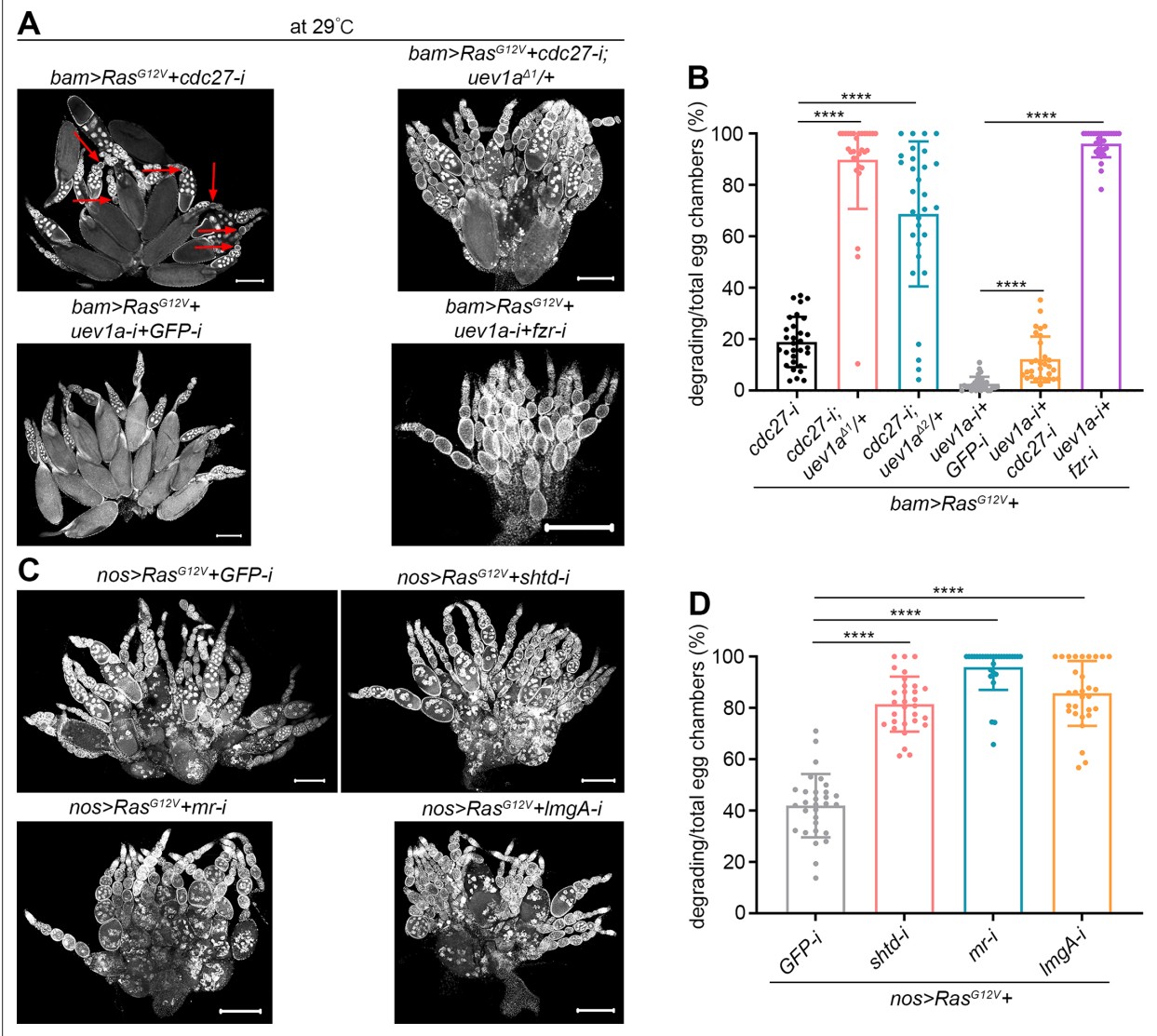

**Figure 5.** Uev1A collaborates with the anaphase-promoting complex or cyclosome (APC/C) complex to mitigate $Ras^{G12V}$-induced nurse cell death. (**A and C**) Representative ovaries (DAPI staining). The red arrows in (**A**) denote degrading egg chambers. Scale bars: 200 μm. (**B and D**) Quantification data. 30 ovaries from 7-day-old (**B**) or 3-day-old (**D**) flies were quantified for each genotype. Statistical significance was determined using one-way ANOVA: **** ($p<0.0001$).

The online version of this article includes the following figure supplement(s) for figure 5:

**Figure supplement 1.** Expression pattern of Uev1A in germarium.

**Figure supplement 2.** Uev1A, Ben, and Cdc27 work together to protect nurse cells from death during normal oogenesis.

Cell division cycle 27 (Cdc27, *Drosophila* APC3) is an essential part of the substrate recognition TPR lobe within the APC/C complex (*Yamano, 2019*). In *bam>RasG^{G12V}+cdc27*-RNAi ovaries, we observed dying nurse cells, a phenotype that was exacerbated with mutations in either *uev1a^{Δ1}* or *uev1a^{Δ2}* (*Figure 5A and B*). Furthermore, knocking down *cdc27* could increase the incidence of dying nurse cells in *bam>RasG^{G12V}+uev1a*-RNAi ovaries (*Figure 5A and B*). Also, we investigated Fizzy-related (Fzr), the *Drosophila* homolog of Cdh1, that is a critical activator of APC/C-dependent proteolysis. It is known that Fzr functions to downregulate mitotic cyclins, including CycA, during cell entry into endocycles (*Sigrist and Lehner, 1997*). Remarkably, nearly all egg chambers in *bam>RasG^{G12V}+uev1a-RNAi+fzr*-RNAi ovaries exhibited degradation (*Figure 5A and B*). Additionally, we knocked down three additional APC/C complex genes in *bam>RasG^{G12V}+uev1a-RNAi* ovaries: *shattered* (*shtd*, encoding *Drosophila* APC1) (*Tanaka-Matakatsu et al., 2007*), *morula* (*mr*, encoding *Drosophila* APC2) (*Kashevsky et al., 2002*), and *lemming A* (*lmgA*, encoding *Drosophila* APC11) (*Nagy et al., 2012*). Among these factors, Shtd is a critical component of the scaffolding platform, while Mr and LmgA are essential components of the catalytic modules within the APC/C complex (*Yamano, 2019*). However, no significant enhancement in nurse cell death was observed, which may be due to the relatively mild phenotype of nurse cell death in *bam>RasG^{G12V}+uev1a-RNAi* ovaries. Therefore, we switched to knocking down these genes in *nos>RasG^{G12V}* ovaries. Notably, knockdown of any of them could significantly exacerbate *Ras^{G12V}*-induced nurse cell death (*Figure 5C and D*), similar to the effect observed with Uev1A downregulation. Collectively, these findings provide genetic evidence that Uev1A collaborates with the APC/C complex to mitigate *Ras^{G12V}*-induced nurse cell death.

Then, we investigated whether Uev1A and the APC/C complex protect nurse cells from death during normal oogenesis. Given that the G2/M-promoting CycA is absent in the G/S-endocycling nurse cells (*Figure 4E*; *Lilly et al., 2000*; *Lilly and Spradling, 1996*), we used the cystocyte-specific driver, bam-GAL4-VP16, to conduct the assays at 29°C. Similar to nurse cells, Uev1A expression was at basal levels in cystocytes, with some expression detected in inner sheath cells and epithelial follicle cells (*Figure 5—figure supplement 1*). Remarkably, nearly all egg chambers in *bam>fzr*-RNAi ovaries exhibited degradation (*Figure 5—figure supplement 2*), underscoring Fzr's critical regulatory role in this context. Uev1A typically partners with Bendless (Ben), the *Drosophila* homolog of mammalian Ubc13 (*Muralidhar and Thomas, 1993*; *Oh et al., 1994*), to perform the E2 enzyme function (*Sancho et al., 1998*; *Zhou et al., 2005*). Ovaries with the *bam>cdc27*-RNAi, *bam>cdc27-RNAi+uev1a-RNAi*, or *bam>cdc27-RNAi+ben*-RNAi genotype showed minimal degradation of egg chambers (*Figure 5—figure supplement 2*). However, degraded egg chambers were prevalent in *bam>cdc27-RNAi+ben-RNAi; uev1a^{Δ1}/++* and *bam>cdc27-RNAi+ben-RNAi; uev1a^{Δ2}/++* ovaries, where Uev1A, Ben, and Cdc27 are all downregulated (*Figure 5—figure supplement 2*). These results suggest that Uev1A, Ben, and the APC/C complex also work together to protect nurse cells from death during normal oogenesis.

## Uev1A and the APC/C complex work together to degrade CycA via the proteasome

The APC/C complex is known to collaborate with the E2 enzymes UBE2C (*Drosophila* homolog: Vihar) and UBE2S (*Drosophila* homolog: Ube2S) to facilitate the proteasomal degradation of CycA during cell cycle progression (*Greil et al., 2022*; *Yamano, 2019*). This degradation involves the assembly of branched ubiquitin chains, incorporating K11, K48, and K63 linkages through a two-step mechanism (*Meyer and Rape, 2014*). However, the involvement of Uev1A in this process remains unexplored. To explore it, we performed biochemical assays in cultured *Drosophila* Schneider 2 (S2) cells, over-expressing tag-fused proteins using *act-GAL4* to enhance expression efficiency. Co-immunoprecipitation (co-IP) assays revealed that Uev1A interacts with several components of the APC/C complex, including Mr (*Drosophila* APC2), Cdc16 (*Drosophila* APC6), and Cdc23 (*Drosophila* APC8). In addition, Cdc27 was found to interact with CycA (*Figure 6A and B*, *Figure 6—figure supplement 1*). Remarkably, RNAi targeting *uev1a*, *ben*, or *cdc27* significantly stabilized CycA proteins after the treatment with cycloheximide (CHX), an inhibitor of protein synthesis (*Figure 6C and D* and *Figure 6—figure supplement 2*). Since the K63 linkage primarily mediates lysosomal degradation via autophagy (*Kwon and Ciechanover, 2017*), we tested CycA stabilization with a chloroquine (CQ, a lysosome inhibitor) or MG132 (a proteasome inhibitor) treatment. Notably, MG132, but not CQ, treatment significantly

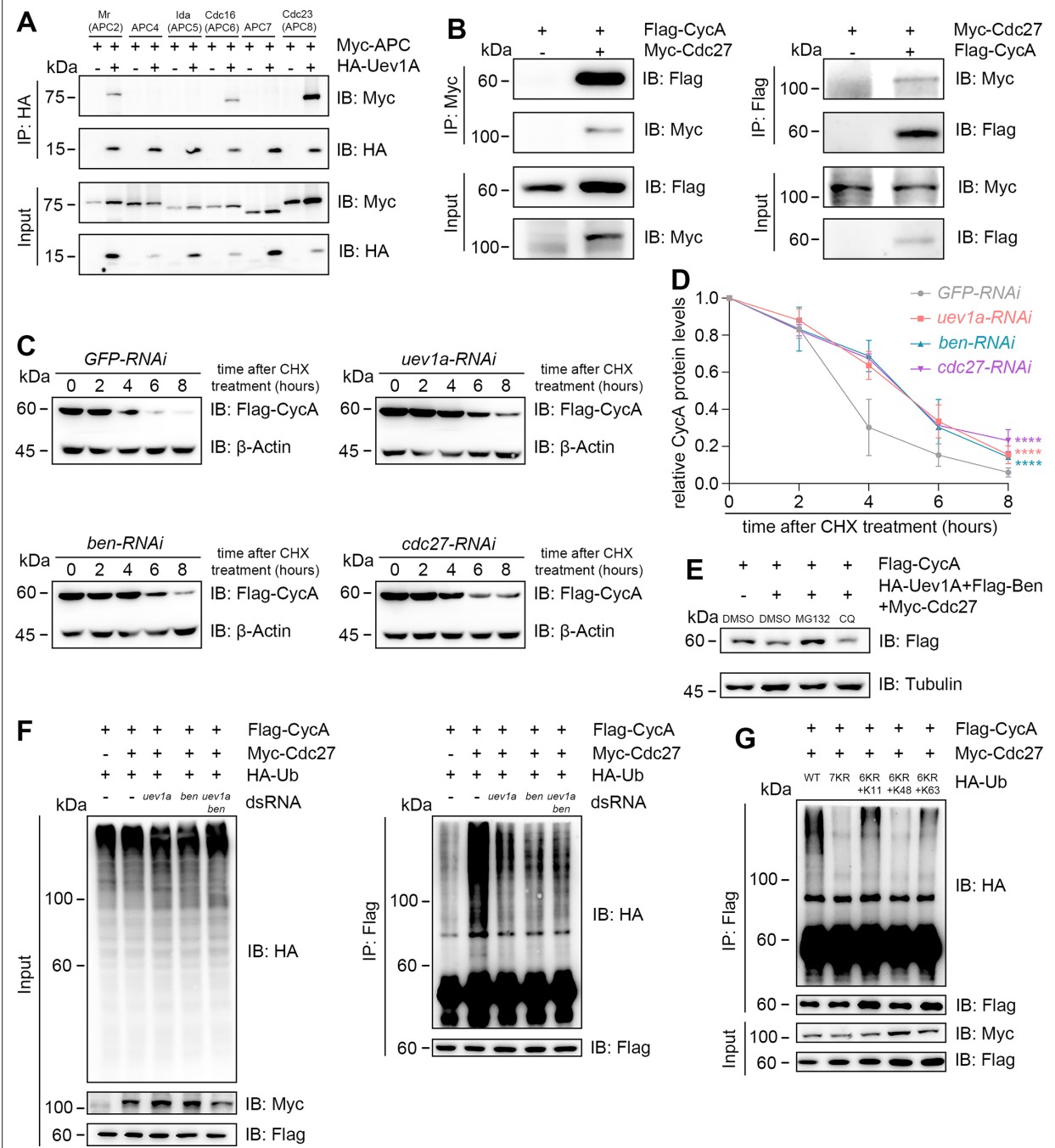

**Figure 6.** Uev1A, Ben, and Cdc27 work together to degrade CycA through the proteasome. (**A and B**) Co-immunoprecipitation (co-IP) assays. The tagged proteins were co-expressed in S2 cells to assess physical interactions. As shown in (**A**), Uev1A interacts specifically with three APC/C subunits: Mr (APC2), Cdc16 (APC6), and Cdc23 (APC8). Assays in (**B**) demonstrate a physical interaction between CycA and Cdc27 (APC3). (**C–E**) CycA stability assays. CHX: a protein-synthesis inhibitor; MG132: a proteasome inhibitor; CQ: a lysosome inhibitor. In (**D**), the relative levels of CycA proteins were quantified using the following formula: (Mean gray value of the CycA/β-Actin band at n hours post-treatment) ÷ (Mean gray value of the CycA/β-Actin band at 0 hr). Three independent replicates were conducted at each time point, and statistical significance was determined using two-way ANOVA with multiple comparisons: **** ($p < 0.0001$). (**F and G**) CycA ubiquitination assays in S2 cells. As shown in (**F**), Cdc27 promotes CycA ubiquitination in a Uev1A/Ben-dependent manner. Assays in (**G**) indicate that the K11 and K63 of ubiquitin are required for CycA ubiquitination.

The online version of this article includes the following source data and figure supplement(s) for figure 6:

**Source data 1.** PDF files that contain original western blots indicating the relevant bands and treatments.

**Source data 2.** Original files for western blot analysis.

*Figure 6 continued on next page*

*Figure 6 continued*

**Figure supplement 1.** Co-immunoprecipitation (co-IP) results.

**Figure supplement 1—source data 1.** PDF files that contain original western blots indicating the relevant bands and treatments.

**Figure supplement 1—source data 2.** Original files for western blot analysis.

**Figure supplement 2.** RNAi efficiency assays.

**Figure supplement 2—source data 1.** PDF files that contain original western blots indicating the relevant bands and treatments.

**Figure supplement 2—source data 2.** Original files for western blot analysis.

stabilized CycA proteins (*Figure 6E*). These results suggest that Uev1A, Ben, and Cdc27 cooperate to promote CycA degradation through the proteasome rather than the lysosome.

To directly validate this, we performed ubiquitination assays for CycA in S2 cells. Cdc27 significantly enhanced CycA ubiquitination, while this effect was markedly reduced upon the knockdown of *uev1a*, *ben*, or both (*Figure 6F*). This result indicates that both Uev1A and Ben are essential for APC/C-mediated proteasomal degradation of CycA. Furthermore, we explored the roles of the seven lysine (K) residues in ubiquitin for polyubiquitin chain formation. The 7KR (lysine to arginine) mutation completely abolished Cdc27-promoted polyubiquitination of CycA. Intriguingly, the 6KR+K48 mutation, in which all K residues except K48 were mutated to R residues, failed to restore polyubiquitination. In contrast, either 6KR+K11 or 6KR+K63 mutation significantly restored polyubiquitination (*Figure 6G*). These findings suggest that the K11 and K63 linkages are primarily responsible for CycA polyubiquitination in *Drosophila* cells. Together with previous studies (*Hofmann and Pickart, 1999*; *McKenna et al., 2001*), we propose that Uev1A and Ben mediate the K63 linkage in this process.

## Uev1A inhibits the overgrowth of *Drosophila* germline tumors driven by oncogenic *Ras^{G12V}*

Given the absence of cell division in normal polyploid nurse cells (*Hammond and Laird, 1985*), their death induced by division-promoting *Ras^{G12V}* represents an artificial stress. Notably, Uev1A was not upregulated in response to this stress (*Figure 3E*), and it executed the function through degrading CycA (*Figures 4–6*). These findings prompted us to investigate whether Uev1A also counteracts oncogenic *Ras*-driven tumorigenesis in diploid cells, which undergo normal cell division. Our prior research demonstrated that oncogenic *Ras^{G12V}* markedly promotes the overgrowth of diploid *bam*-deficient germline tumors (*Zhang et al., 2024b*), which are highly resistant to cell death (*Zhang et al., 2023*; *Zhao et al., 2018*). Intriguingly, knocking down *uev1a* significantly enhanced the overgrowth of these tumors, while overexpressing Uev1A suppressed it (*Figure 7A and B*). These results indicate that Uev1A also plays a role in counteracting oncogenic *Ras*-driven tumorigenesis in diploid cells.

Considering the therapeutic potential of gene upregulation in inhibiting tumor growth, a critical concern is its impact on normal physiological processes. In this study, we examined the impact of Uev1A overexpression on *Drosophila* oogenesis and GSC maintenance. Notably, the *nos>Uev1A* flies remained fertile, and their ovaries appeared morphologically similar to those of the *nos>lacZ* control flies (*Figure 7C*), indicating normal oogenesis. Furthermore, each germarium in the ovaries of 14-day-old *nos>Uev1A* flies contained a similar number of GSCs as those in the *nos>lacZ* control flies (*Figure 7D and E*). These results suggest that Uev1A overexpression does not disrupt normal oogenesis and GSC maintenance.

## UBE2V1 and UBE2V2 inhibit the overgrowth of human colorectal tumors driven by oncogenic *KRAS*

Our findings in *Drosophila* prompted us to explore the tumor-suppressive effects of UBE2V1 and UBE2V2 on the growth of *RAS*-mutant human tumors. Using the Kaplan-Meier plotter (**https://kmplot.com/analysis**), we first evaluated the correlation between UBE2V1/2 expression and prognosis in several types of *RAS*-mutant cancer patients, including melanoma, myeloma, lung cancer, and colorectal cancer. Among them, higher expression levels of UBE2V1/2 were significantly associated with improved relapse-free survival in *KRAS*-mutant colorectal cancer patients (*Figure 8A*). RNA-seq data from The Cancer Genome Atlas (TCGA) showed that *UBE2V1* and *UBE2V2* are transcribed at similar levels in colorectal cancer patients with oncogenic *RAS* mutations (including *KRAS*, *HRAS*, and

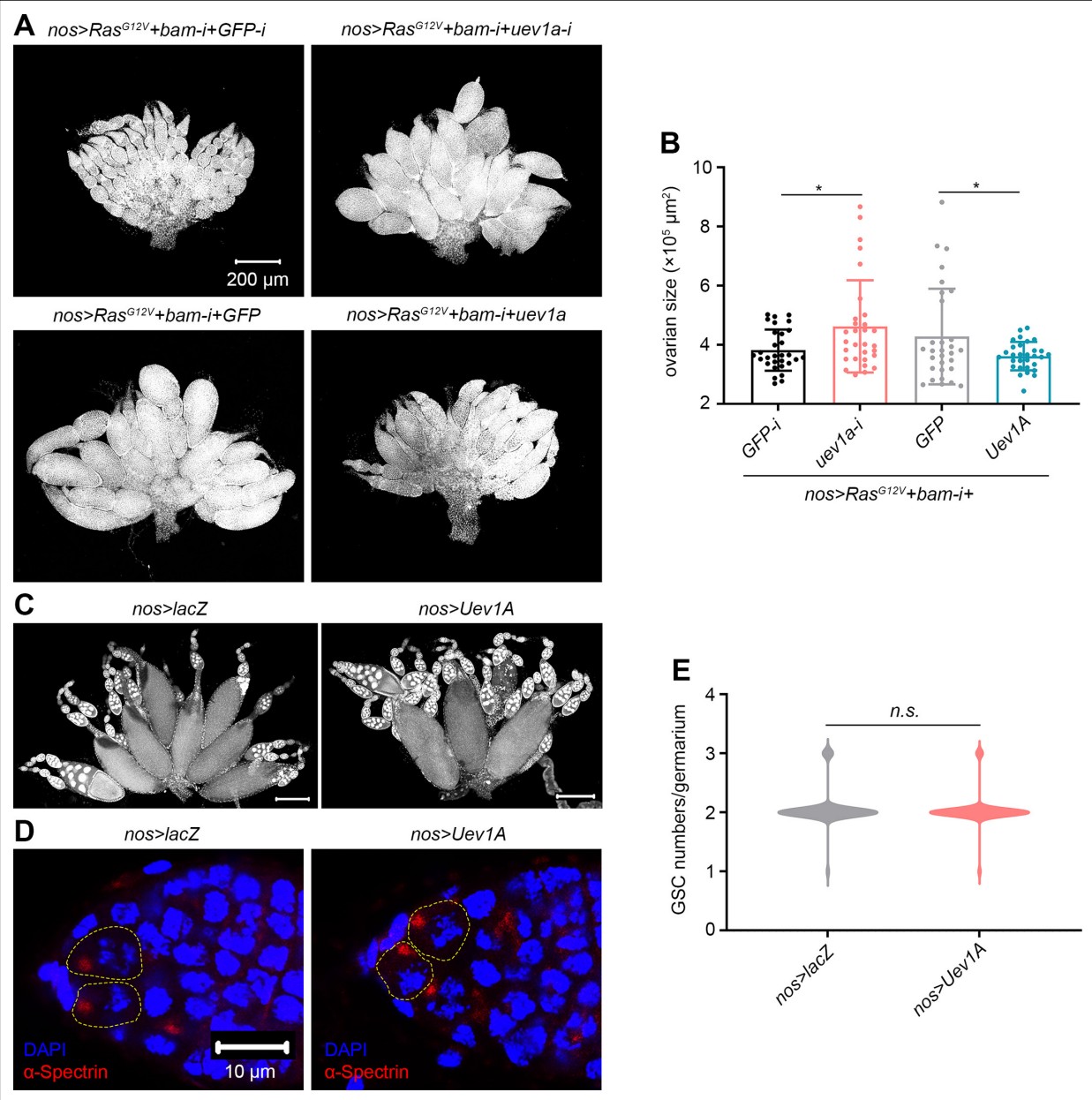

**Figure 7.** Uev1A inhibits the overgrowth of germline tumors induced by oncogenic *Ras^{G12V}*. (**A and C**) Representative ovaries (DAPI staining). All images in (**A**) are of the same magnification. Scar bars in (**C**): 200 µm. (**B**) Quantification data for ovarian size. The largest 2D area of each ovary in a single confocal focal plane was scanned, and its size was measured using ImageJ. 30 ovaries from 3-day-old flies were analyzed for each genotype. (**D**) Representative samples. The germline stem cells (GSCs) within stem cell niches are outlined by yellow dashed lines. Both images are of the same magnification. (**E**) Quantification data for GSC numbers per germarium. Germ cells that directly contact cap cells and contain dot-like spectrosomes were counted as GSCs. 100 germaria from 14-day-old flies were quantified for each genotype. In (**B and E**), statistical significance was determined using t test: *n.s.* (p>0.05) and * (p<0.05).

*NRAS*) as in those without such mutations (***Figure 8—figure supplement 1***). These findings suggest that UBE2V1 and UBE2V2 are not upregulated in response to oncogenic *RAS* mutations, paralleling the behavior of Uev1A in *Drosophila* ovarian nurse cells (***Figure 3E***).

To investigate the potential tumor-suppressive roles of UBE2V1 and UBE2V2 in *KRAS*-mutant colorectal cancer, we performed individual knockdowns of each gene in two human colorectal cancer cell lines: SW480 cells (carrying the *KRAS^{G12V}* mutation) and HCT116 cells (carrying the *KRAS^{G13D}* mutation). Notably, individual knockdown of either *UBE2V1* or *UBE2V2* only mildly affected the proliferation of SW480 cells (***Figure 8—figure supplement 2***), suggesting potential functional redundancy

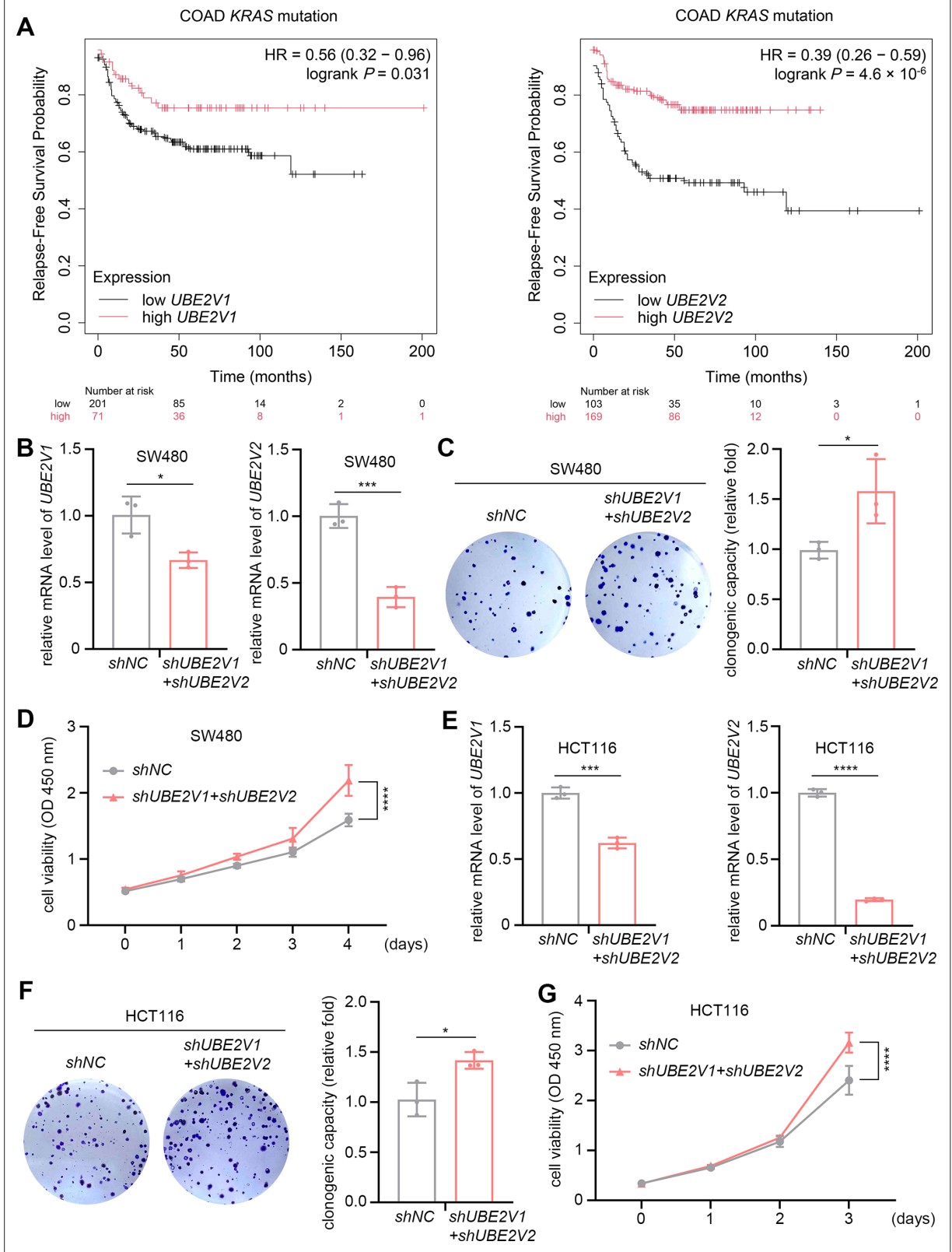

**Figure 8.** Prognostic significance and tumor-suppressive effects of UBE2V1 and UBE2V2 on *KRAS*-mutant colorectal cancer. (**A**) Kaplan-Meier analysis of relapse-free survival in *KRAS*-mutant colorectal cancer patients with high or low expression levels of UBE2V1 and UBE2V2. (**B and E**) The knockdown efficiency assays. The relative mRNA levels were normalized to *GAPDH*. (**C–G**) Assays to evaluate the effects of *UBE2V1-* and *UBE2V2-RNAi* on colony formation and cell viability in SW480 and HCT116 cells. In (**B, C, E, and F**), three independent replicates were conducted, and statistical significance

*Figure 8 continued on next page*

*Figure 8 continued*

was determined using t test. In (**D and G**), five (**D**) and six (**G**) independent replicates were conducted at each time point, and statistical significance was determined using two-way ANOVA with multiple comparisons. * (p<0.05), *** (p<0.001), and **** (p<0.0001).

The online version of this article includes the following figure supplement(s) for figure 8:

**Figure supplement 1.** The Cancer Genome Atlas (TCGA) analysis comparing UBE2V1 and UBE2V2 expression levels in colorectal cancer patients with and without *RAS* mutations.

**Figure supplement 2.** Knocking down either *UBE2V1* or *UBE2V2* alone mildly influences the growth of colorectal cancer cell lines.

between the two proteins. However, combined knockdown of *UBE2V1* and *UBE2V2* significantly enhanced colony formation and cell viability in both SW480 and HCT116 cells, as demonstrated by clonogenic and CCK8 assays (*Figure 8B–G*). These results indicate that UBE2V1 and UBE2V2 exert tumor-suppressive effects in colorectal cancer cells harboring oncogenic *KRAS* mutations.

Next, we examined whether overexpression of UBE2V1 or UBE2V2 could suppress the growth of SW480 and HCT116 cells. The overexpression efficiency of each protein was confirmed by western blotting (*Figure 9—figure supplement 1A*). Using 5-ethynyl-2'-deoxyuridine (EdU) incorporation assays, we found that overexpression of either UBE2V1 or UBE2V2 significantly inhibited the proliferation of both cancer cell lines in vitro (*Figure 9—figure supplement 2A and B*). This antiproliferative effect was further supported by clonogenic and cell viability assays (*Figure 9—figure supplement 2C–E*).

To validate the tumor-suppressive effects of UBE2V1 and UBE2V2 in vivo, we established stable SW480 cell lines overexpressing each protein, with overexpression confirmed by western blotting (*Figure 9—figure supplement 1B*). In subcutaneous tumorigenesis assays using Balb/c nude mice, overexpression of either UBE2V1 or UBE2V2 significantly inhibited tumor formation compared to control (*Figure 9A and B*). Immunohistochemical analysis of the resulting tumors showed a marked decrease in Cyclin A expression and Ki-67-positive cells, indicating reduced proliferation (*Figure 9C and D*). Notably, the tumor-suppressive effect of UBE2V1 was more robust than that of UBE2V2, consistent with the stronger protective effect of UBE2V1 against Ras$^{G12V}$-induced nurse cell death in *Drosophila* (*Figure 2F and G*). Taken together, these findings provide strong evidence for the tumor-suppressive roles of UBE2V1 and UBE2V2 in KRAS-mutant colorectal cancer.

## Discussion

It is well established that oncogenic Ras can induce DNA replication stress, leading to cellular senescence or death (*Hills and Diffley, 2014*; *Kotsantis et al., 2018*). However, our previous (*Zhang et al., 2024b*) and current findings demonstrated that oncogenic *Ras$^{G12V}$* can also trigger cell death in polyploid *Drosophila* ovarian nurse cells through aberrantly promoting their division. In this study, we performed a genome-wide genetic screen and identified the E2 enzyme Uev1A as a crucial protector against this specific form of cellular stress. Mechanistically, Uev1A collaborates with the APC/C complex (E3) to facilitate the proteasomal degradation of CycA, which overexpression alone can also trigger nurse cell death. Furthermore, Uev1A and its human homologs, UBE2V1 and UBE2V2, counteract oncogenic *Ras*-driven tumorigenesis in diploid *Drosophila* germline and human colorectal tumor cells, respectively. These findings highlight the critical role of Uev1A in counteracting oncogenic *Ras* stimuli in both polyploid and diploid cells (*Figure 9E*).

Our studies reveal that the DDR pathway protects against *Ras$^{G12V}$*-induced nurse cell death (*Figure 3A and B*), suggesting that this cellular stress both activates and is subsequently suppressed by the DDR. In contrast, p53 promotes nurse cell death under the same conditions (*Figure 3A and B*). Interestingly, previous research has also demonstrated distinct roles of p53 and Lok, a crucial DDR regulator, in regulating nurse cell death during mid-oogenesis. While Lok overexpression triggers nurse cell death, p53 overexpression does not. Furthermore, Lok-induced nurse cell death remains unaffected by *p53* mutation, indicating that its mechanism operates independently of p53 (*Bakhrat et al., 2010*). The underlying mechanisms driving these differences warrant further investigation.

Of note, the suppressive effects of Uev1A on *Ras$^{G12V}$*; *bam$^{RNAi}$* germline tumors in *Drosophila* were less pronounced than that of UBE2V1/2 on oncogenic *KRAS$^{G12V}$*-driven human colorectal tumor xenografts in nude mice (compare *Figure 7A and B* with *Figure 9A and B*). Our previous research has shown that *Ras$^{G12V}$*; *bam$^{-/-}$* germline tumor cells divide infrequently (*Zhang et al., 2024b*). This

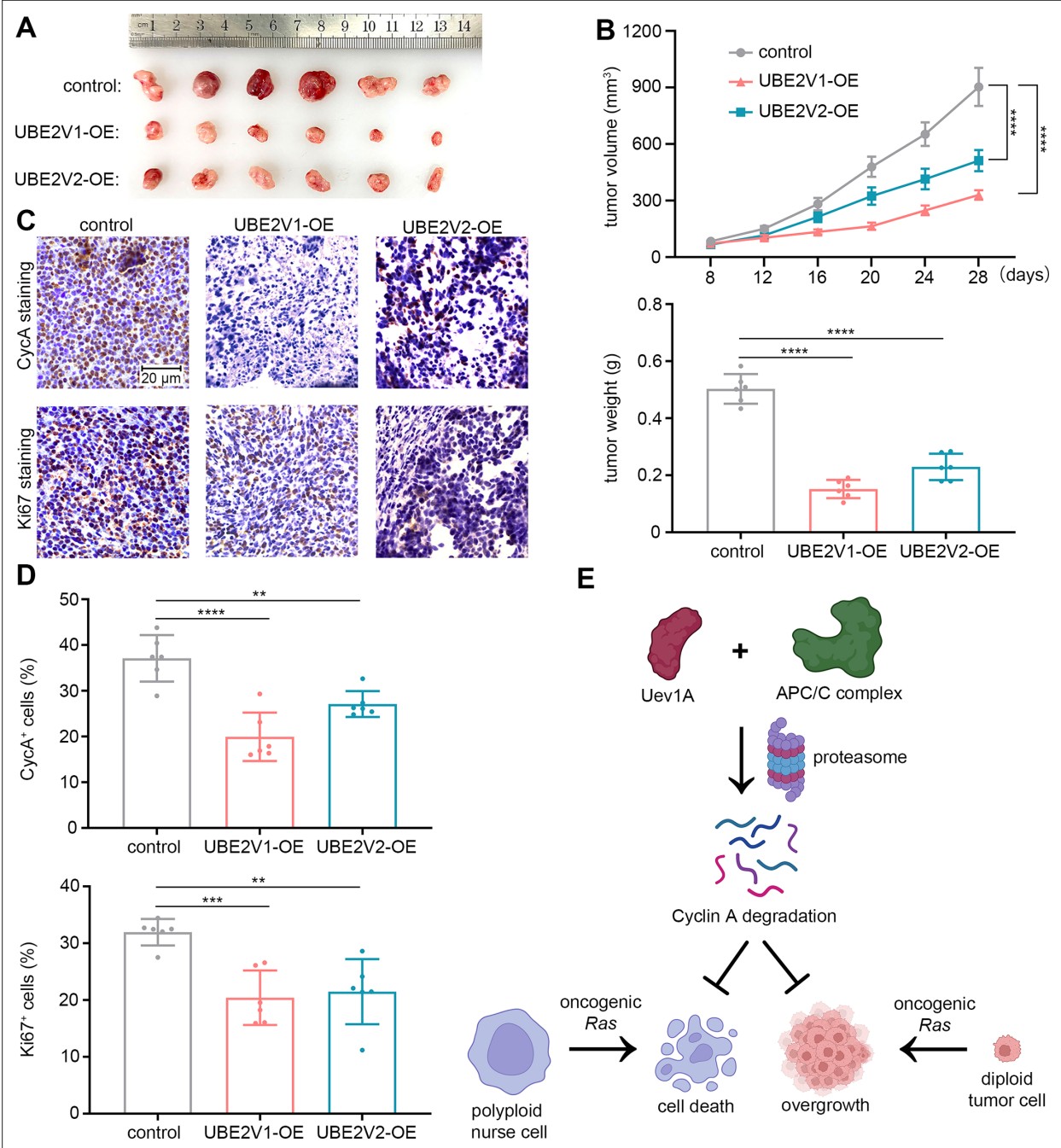

**Figure 9.** Overexpression of UBE2V1 or UBE2V2 suppresses the growth of *KRAS*-mutant colorectal cancer. (**A and B**) Subcutaneous tumorigenesis assays in nude mice, where tumors were excised, photographed, and weighed 28 days after tumor cell injection. (**C and D**) Immunohistochemical staining to assess CycA expression and Ki-67 positivity in tumor tissues. All images in (**C**) are of the same magnification. In (B-1), six independent replicates were conducted, and statistical significance was determined using two-way ANOVA with multiple comparisons. In (B-2 and D), six independent replicates were conducted, and statistical significance was determined using one-way ANOVA. ** ($p<0.01$), *** ($p<0.001$), and **** ($p<0.0001$). (**E**) Working model. By degrading CycA, Uev1A and the E3 APC/C complex counteract oncogenic *Ras* stimuli, thereby protecting against cell death in polyploid *Drosophila* nurse cells and suppressing overgrowth in diploid *Drosophila* germline and human colorectal tumor cells.

The online version of this article includes the following source data and figure supplement(s) for figure 9:

**Figure supplement 1.** Validation of UBE2V1/2 overexpression in colorectal cancer cell lines.

**Figure supplement 1—source data 1.** PDF files that contain original western blots indicating the relevant bands and treatments.

**Figure supplement 1—source data 2.** Original files for western blot analysis.

**Figure supplement 2.** UBE2V1/2 overexpression suppresses the growth of colorectal cancer cell lines.

suggests that the relatively weak suppression observed in these tumors may be due to the fact that Uev1A's mechanism of action is cell cycle-dependent: the more frequently tumor cells divide, the more effectively Uev1A can suppress tumor overgrowth. We did not observe significant negative effects of Uev1A overexpression on GSC maintenance in *Drosophila* ovaries (*Figure 7D and E*). This is likely because GSCs also divide infrequently (*Morris and Spradling, 2011*). Therefore, these findings underscore the importance of considering the potential negative effects of Uev1A/UBE2V1/2 overexpression in other contexts, particularly when cells divide rapidly.

RAS oncoproteins have long been considered 'undruggable', due to the lack of deep-binding, targetable pockets (*Moore et al., 2020*). To date, targeted therapies have been approved only for KRAS$^{G12C}$ (*Canon et al., 2019*; *Ou et al., 2022*; *Skoulidis et al., 2021*). Recently, small-molecule pan-KRAS degraders, developed using the proteolysis-targeting chimeras (PROTACs) strategy, have shown promise (*Popow et al., 2024*), though their clinical effectiveness and safety remain to be fully validated. Alternatively, targeting key regulators in the RAS signaling pathway—such as SOS, SHP2, Farnesyltransferase, Raf, MEK, ERK, and PI3K—has emerged as an attractive therapeutic approach (*Moore et al., 2020*). Recent research has also explored the strategy of hyperactivating oncogenic *RAS* to trigger cell death as a potential means to suppress the overgrowth of human colon tumor xenografts in nude mice (*Dias et al., 2024*). Additionally, targeting the cellular dependencies driven by oncogenic *RAS* may trigger RAS-specific synthetic lethality, presenting a promising therapeutic approach (*Moore et al., 2020*). Our findings highlight the tumor-suppressive effects of UBE2V1 and UBE2V2 on human colorectal tumors driven by oncogenic *KRAS* (*Figures 8 and 9*). Notably, both of these two E2 enzymes are relatively small, with UBE2V1 consisting of 147 amino acid residues and UBE2V2 145 residues (*Figure 2E*). This raises the exciting possibility that their upregulation, through mRNA delivery or small-molecule agonists, could offer a promising therapeutic approach for human cancer.

# Materials and methods

## Fly husbandry
Cross experiments were conducted at 25°C, except for some performed at 29°C as noted.

## Transgenic flies
### UASz-flag-Ras$^{G12V}$, UASz-uev1a, UASz-UBE2V1, and UASz-UBE2V2
The coding sequences (CDSs) of *flag-Ras$^{G12V}$*, *uev1a-RA*, *UBE2V1*, and *UBE2V2* were cloned into the *pUASz1.0* vector. These plasmids were then microinjected into fertilized fly embryos with the genotype '*nos-int; attP40*', resulting in the generation of transgenic fly strains. The DNA sequence encoding 3xFlag was as follows: ATGGACTACAAAGACCATGACGGTGATTATAAAGATCATGACAT CGATTACAAGGATGACGATGACAAGCTT.

### uev1a$^{Δ1}$ and uev1a$^{Δ2}$ mutants
The gRNA targeting the second coding exon of the *uev1a* gene was designed with the DNA sequence 'GATCCAGCTCCTCCAGTAAGCGG' and cloned into the *pCFD3* vector. The plasmids were then microinjected into fertilized fly embryos with the genotype '*nos-int; attP40*' to generate transgenic fly strain. These transgenic flies were subsequently crossed with '*nos-cas9; FRT2A*' flies to generate *uev1a* knockout mutants, whose molecular information was identified by Sanger sequencing.

### uev1a-flag knock-in
The gRNA targeting the second-to-last coding exon of the *uev1a* gene was designed with the DNA sequence 'GTTATAAACCGGAGCGTTGGTGG' and cloned into the *pU6-BbsI-chiRNA* vector. The homology arms consisted of 991 bp upstream of the stop codon ('TAG'), followed by a 3xFlag tag sequence, the 'TAG' stop codon, and 979 bp downstream of 'TAG'. To prevent cleavage by the gRNA, the targeting sequence within the upstream homology arm was mutated to 'CCTACCCTGCGCTTCA TTA', ensuring that the amino acid sequence remained unaltered. These DNA sequences were synthesized into the *pUASz1.0* vector between the *BamHI* and *KpnI* sites and used as a double-stranded DNA

(dsDNA) donor for homologous recombination-mediated repair. The *pU6-uev1a-gRNA-2* (100 ng/µL) and *pUASz-uev1a-donor* (100 ng/µL) plasmids were then co-injected into fertilized fly embryos with the genotype '*nos-cas9 (on X)*' (**Kondo and Ueda, 2013**). The resulting *uev1a-flag* knock-in fly strain was identified by PCR and confirmed by Sanger sequencing.

## Immunofluorescent staining

Fly ovaries were dissected in the PBS solution, fixed by the 4% paraformaldehyde solution (diluted in PBS) for 3 hr, washed by the PBST solution (0.3% Triton X-100 diluted in PBS) for 1 hr at room temperature (RT), then incubated with the primary antibodies (diluted in PBST) overnight at 4°C, washed by the PBST solution for 1 hr at RT, incubated with the secondary antibodies and 0.1 µg/mL DAPI (diluted in PBST) overnight at 4°C, washed by the PBST solution for 1 hr at RT, and subsequently mounted using 70% glycerol (autoclaved).

## Construction of plasmids used in S2 cells

The CDSs of the genes were amplified from the cDNAs derived from either S2 cells or adult wild-type (*w1118*) flies and subsequently cloned into the *pUASt-attB* vector. N- or C-terminal tags were incorporated based on prior studies or structural predictions from AlphaFold. The following proteins were tagged at the N-terminus: APC4, Ben, CDC16, CDC23, Cdc27, CycA, Fzr, Fzy, Ida, Mr, Ub, $Ub^{7KR}$, $Ub^{6KR+K11}$, $Ub^{6KR+K48}$, $Ub^{6KR+K63}$, and Uev1A. In contrast, APC7 was tagged at the C-terminus.

## Culture and transfection of S2 cells

S2 cells were cultured in insect medium with 10% FBS and incubated at 27°C without $CO_2$. To transfect, cells were plated in six-well plates to reach 70% confluence and incubated for 3 hr at 27°C. Then, a pre-complexed mixture of plasmid DNA and X-tremeGENE HP DNA transfection reagent was added slowly to the cultures. The transfection process followed the manufacturer's instructions. After gentle mixing, cells were incubated at 27°C, and samples were collected 36 hr after transfection.

## Immunoprecipitation, immunoblotting, ubiquitination, and protein stability assays in S2 cells

### Immunoprecipitation assay

The transfected S2 cells were lysed using NP-40 lysis buffer supplemented with protease inhibitors at 4°C for 30 min. The supernatants were incubated with primary antibodies at 4°C for 3 hr, followed by incubation with Protein G Sepharose for 2 hr. The beads were washed five times with NP-40 lysis buffer and boiled for 5 min in 2× SDS protein loading buffer (0.25 M Tris-HCl [pH 6.8], 78 mg/mL DTT, 100 mg/mL SDS, 50% glycerol, and 5 mg/mL bromophenol blue). Then the samples were subjected to western blotting.

### Immunoblotting assay

After 36 hr of transfection, the S2 cells were harvested and lysed in 2× SDS protein loading buffer. The lysates were then boiled for 5 min and subjected to western blotting.

### Ubiquitination assay

S2 cells were transiently transfected with the indicated plasmid combinations. 30 hr post-transfection, cells were treated with 50 µM MG132 for 6 hr. Proteins were then immunoprecipitated and analyzed by western blotting.

### Protein stability assay

S2 cells were treated with 20 µg/mL of the protein synthesis inhibitor CHX for specified time intervals prior to harvesting.

## RNAi assay in S2 cells

Two double-stranded RNAs (dsRNAs) targeting distinct regions of the relevant genes were synthesized using the T7 RiboMAX Express RNAi System. S2 cells were seeded in six-well tissue culture plates and treated with 15 µg of dsRNA in the culture medium, followed by incubation for 24 hr.

Expression plasmids were then transiently transfected into the cells, after which an additional 10 µg of dsRNA was added. The cells were cultured for another 36 hr before harvesting. As a negative control, dsRNA targeting the *AcGFP* gene was used. Templates for dsRNA synthesis were generated by PCR amplification of S2 cell genomic DNA using the primers listed in the Key resources table.

### Human cell culture

The human colon cancer cell lines SW480 and HCT116, as well as the human embryonic kidney cell line 293T, were purchased from the American Type Culture Collection (ATCC), authenticated by STR profiling, and tested negative for mycoplasma contamination. These cell lines were cultured in Dulbecco's Modified Eagle Medium supplemented with 10% FBS, and the cultures were maintained at 37°C with 5% $CO_2$ in a humidified incubator.

### Gene knockdown assay

Short hairpin RNA (shRNA) constructs were generated using the lentiviral vector *pLKO-CMV-copGFP-puro*. Lentiviral particles were produced by co-transfecting each shRNA plasmid together with the packaging plasmids *psPAX2* and *pMD2.G* into 293T cells using Lipofectamine 2000, following the manufacturer's protocol. SW480 and HCT116 cells were subsequently transduced with the harvested lentiviral supernatants to establish stable cell lines expressing *shNC* (negative control), *shUBE2V1*, or *shUBE2V2*. The shRNA sequences were included in the Key resources table.

### Overexpression assay

The lentiviral vector *pCDH-CMV* was used for gene overexpression, while *pMD2.G* and *psPAX2* served as lentiviral packaging plasmids. Target plasmids, *pCDH-CMV-UBE2V1* and *pCDH-CMV-UBE2V2*, were synthesized by the GENEWIZ company (Suzhou, China). Plasmids were transfected into 293T cells using Lipofectamine 2000 to generate lentiviral particles, according to the manufacturer's protocol. SW480 and HCT116 cells were then transduced with these lentiviruses to establish stable cell lines expressing UBE2V1 and UBE2V2, respectively. To select for stably transduced cells, the cultures were maintained in medium containing 2 µg/mL puromycin.

### Quantitative real-time PCR

Total RNA was extracted using the TriQuick Reagent kit and reverse-transcribed into cDNA using the SPARKscript II All-in-one RT SuperMix kit. Quantitative real-time PCR was performed using the SYBR Green Premix Pro Taq HS qPCR kit. *GAPDH* was used as the endogenous control for normalization. The relative mRNA levels of target genes were determined using the $2^{-\Delta\Delta CT}$ method. The primer sequences were listed in the Key resources table.

### EdU incorporation assay

Cells were seeded in 24-well plates at a density of 100,000 cells per well. Cell proliferation was assessed using the EdU incorporation assay kit, following the manufacturer's instructions. Briefly, EdU reagent was added to the culture medium, and cells were incubated for 2 hr to label DNA-synthesizing cells. After incubation, the medium containing EdU was removed, and cells were washed with PBS, fixed with fixative solution for 10 min, and then subjected to a click reaction for fluorescence labeling. Fluorescence microscopy was used to capture images and calculate the percentage of EdU$^+$ cells.

### Colony formation assay

Cells were seeded in six-well plates at a density of 500 cells per well and cultured at 37°C in a 5% $CO_2$ incubator for 10 days without medium change. After 10 days, cells were gently washed with PBS to remove non-adherent cells, then stained with 0.5% crystal violet solution (containing 10% methanol and 1% acetic acid) for 15 min. The plates were subsequently rinsed with water to remove excess stain and air-dried. Colonies were photographed and counted using image analysis software.

### CCK8 cell viability assay

Cells were seeded in 96-well plates at a density of 2000 cells per well. Cell viability was assessed over 5 days using a CCK8 kit following the manufacturer's instructions. Measurements were taken on day 0 (baseline) and daily from day 1 to day 4. At each time point, 10 µL of CCK8 solution was added to each

well, and the plates were incubated for 2 hr at 37°C in a humidified incubator. Absorbance at 450 nm was measured using a microplate reader to assess cell viability and proliferation dynamics throughout the experimental period.

## Western blotting

Cells were lysed with RIPA buffer containing protease inhibitors, and protein concentrations were determined using a BCA Protein Assay Kit. Protein samples (20 µg per lane) were separated by SDS-PAGE and transferred to PVDF membranes. Membranes were blocked with 5% non-fat milk and incubated overnight at 4°C with primary antibodies. Following washes, membranes were incubated with HRP-conjugated secondary antibodies and developed using ECL. Band intensities were quantified by densitometry.

## Immunohistochemistry assay

Tissue sections were deparaffinized with xylene, rehydrated through a graded ethanol series, and subjected to antigen retrieval in citrate buffer (pH 6.0) at 98°C for 10 min. Endogenous peroxidase activity was blocked using 3% hydrogen peroxide, and nonspecific binding was blocked with 5% BSA. Sections were incubated overnight at 4°C with primary antibodies. After incubation with HRP-conjugated secondary antibodies, sections were developed using DAB, counterstained with hematoxylin, and mounted. Images were captured using a light microscope.

## Animal experiment

Male Balb/c nude mice (7-week-old) were used to assess the tumorigenic potential of SW480 colon cancer cells. The mice were housed under specific pathogen-free conditions with a 12 hr light/dark cycle and had ad libitum access to food and water. SW480 cells were stably transfected with either the empty vector *pCDH-CMV* (control), *pCDH-CMV-UBE2V1* (UBE2V1-OE), or *pCDH-CMV-UBE2V2* (UBE2V2-OE), and $5 \times 10^6$ cells were subcutaneously injected into the right flank of each mouse. Each group consisted of six mice. Tumor growth was monitored approximately 1 week after injection. Tumor volumes were measured every 4 days using calipers and calculated using the formula: volume = length × width$^2$/2. The experiment was terminated when tumors reached ~1000 mm$^3$. The mouse experiments in this study were approved by the Institutional Animal Care and Use Committee at Nankai University (Approval Number: 2025-SYDWLL-000588). All animal procedures were conducted in compliance with the committee's protocols and under its supervision. Mice were humanely euthanized in accordance with institutional and national ethical guidelines.

## Image collection and processing

Fluorescent images were captured using a Zeiss LSM 710 confocal microscope (Carl Zeiss AG, BaWü, GER) and processed with Adobe Photoshop 2022 (San Jose, CA, USA), ImageJ (NIH, Bethesda, MD, USA), and ZEN 3.0 SR imaging software (Carl Zeiss).

# Acknowledgements

We gratefully thank Eric H Baehrecke, Michael Buszczak, Zheng Guo, Yuu Kimata, Ruth Lehmann, Erika Matunis, Addgene, ATCC, BDSC, CEMCS, DGRC, GenBank, and THFC for providing antibodies, plasmids, cell lines, and fly strains. This study was supported by National Natural Science Foundation of China (NSFC) grants to Shaowei Zhao (32270841, 32070871), Shian Wu (32170714), and Hongru Zhang (32400759), as well as by a Natural Science Foundation of Tianjin grant (S24ZDD020) to Hongru Zhang.

## Additional information

### Funding

| Funder | Grant reference number | Author |
| --- | --- | --- |
| National Natural Science Foundation of China | 32270841 | Shaowei Zhao |
| National Natural Science Foundation of China | 32070871 | Shaowei Zhao |
| National Natural Science Foundation of China | 32170714 | Shian Wu |
| National Natural Science Foundation of China | 32400759 | Hongru Zhang |
| Natural Science Foundation of Tianjin | S24ZDD020 | Hongru Zhang |

The funders had no role in study design, data collection and interpretation, or the decision to submit the work for publication.

### Author contributions

Qi Zhang, Yunfeng Wang, Xueli Fu, Ziguang Wang, Data curation, Software, Formal analysis, Validation, Investigation, Methodology, Writing – review and editing; Yang Zhang, Muhan Yang, Ruixing Zhang, Data curation, Formal analysis, Validation, Investigation, Writing – review and editing; Lizhong Yan, Yuejia Wang, Dongze Song, Data curation, Formal analysis, Validation, Investigation, Methodology, Writing – review and editing; Hongru Zhang, Data curation, Formal analysis, Funding acquisition, Investigation, Methodology, Project administration, Software, Supervision, Validation, Visualization, Writing – review and editing; Shian Wu, Data curation, Software, Formal analysis, Supervision, Funding acquisition, Validation, Investigation, Visualization, Methodology, Project administration, Writing – review and editing; Shaowei Zhao, Conceptualization, Resources, Data curation, Software, Formal analysis, Supervision, Funding acquisition, Validation, Investigation, Visualization, Methodology, Writing – original draft, Project administration, Writing – review and editing

### Author ORCIDs

Hongru Zhang ⓘ https://orcid.org/0009-0000-7447-8750
Shian Wu ⓘ https://orcid.org/0000-0003-4990-6594
Shaowei Zhao ⓘ https://orcid.org/0000-0002-4544-7215

### Ethics

The mouse experiments in this study were approved by the Institutional Animal Care and Use Committee at Nankai University (Approval Number: 2025-SYDWLL-000588). All animal procedures were conducted in compliance with the committee's protocols and under its supervision. Mice were humanely euthanized in accordance with institutional and national ethical guidelines.

Reviewer #1 (Public review): https://doi.org/10.7554/eLife.107104.3.sa1
Reviewer #2 (Public review): https://doi.org/10.7554/eLife.107104.3.sa2
Author response https://doi.org/10.7554/eLife.107104.3.sa3

---

## Additional files

### Supplementary files

MDAR checklist

Source data 1. Screen results.

Source data 2. All genotypes.

Source data 3. Raw quantification data.

## Data availability

The results of the deficiency screening are provided in *Source data 1*. All genotypes are listed in *Source data 2*, and the raw quantification data can be found in *Source data 3*.

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

# Appendix 1

## Appendix 1—key resources table

| Reagent type (species) or resource | Designation | Source or reference | Identifiers | Additional information |
|---|---|---|---|---|
| Strain, strain background (*Mus musculus*) | Balb/c (CAnN.Cg-Foxn1$^{nu}$/Crl) nude mice | Beijing Vital River Laboratory Animal Technology Co., Ltd. | 401 | |
| Genetic reagent (*Drosophila melanogaster*) | bam-GAL4-VP16 | *Chen and McKearin, 2003* | | |
| Genetic reagent (*D. melanogaster*) | nos-GAL4-VP16 | *Van Doren et al., 1998* | | |
| Genetic reagent (*D. melanogaster*) | nos-cas9 | *Kondo and Ueda, 2013* | | |
| Genetic reagent (*D. melanogaster*) | FRT2A | BDSC | 1997 | |
| Genetic reagent (*D. melanogaster*) | puc-lacZ | Bloomington Drosophila Stock Center (BDSC) | 98329 | |
| Genetic reagent (*D. melanogaster*) | UAS-bam-RNAi | BDSC | 33631 | |
| Genetic reagent (*D. melanogaster*) | UASp-BicD-RNAi | TsingHua Fly Center (THFC) | THU4454 | |
| Genetic reagent (*D. melanogaster*) | UASp-cdc27-RNAi | THFC | TH201500102.S | |
| Genetic reagent (*D. melanogaster*) | UASp-Cdk1 | BDSC | 65396 | |
| Genetic reagent (*D. melanogaster*) | UASp-CycA | BDSC | 85308 | |
| Genetic reagent (*D. melanogaster*) | UASp-CycB | BDSC | 85312 | |
| Genetic reagent (*D. melanogaster*) | UASp-dsor1-RNAi | THFC | THU0677 | |
| Genetic reagent (*D. melanogaster*) | UASp-fzr-RNAi | THFC | TH201500745.S | |
| Genetic reagent (*D. melanogaster*) | UASp-GFP | *Zhang et al., 2024a* | | |
| Genetic reagent (*D. melanogaster*) | UASp-GFP-RNAi | BDSC | 44412, 44415 | |
| Genetic reagent (*D. melanogaster*) | UASz-lacZ | *Zhang et al., 2024b* | | |
| Genetic reagent (*D. melanogaster*) | UASp-lmgA-RNAi | THFC | THU4085 | |
| Genetic reagent (*D. melanogaster*) | UASp-lok-RNAi | THFC | TH01867.N | |
| Genetic reagent (*D. melanogaster*) | UASp-mr-RNAi | THFC | THU5250 | |
| Genetic reagent (*D. melanogaster*) | UASp-p53-RNAi | THFC | THU5318 | |
| Genetic reagent (*D. melanogaster*) | UASp-Ras$^{G12V}$ | *Zhang et al., 2024b* | | |
| Genetic reagent (*D. melanogaster*) | UASp-rl-RNAi | THFC | THU3530 | |
| Genetic reagent (*D. melanogaster*) | UASp-shtd-RNAi | THFC | TH201500835.S | |

*Appendix 1 Continued on next page*

*Appendix 1 Continued*

| Reagent type (species) or resource | Designation | Source or reference | Identifiers | Additional information |
|---|---|---|---|---|
| Genetic reagent (*D. melanogaster*) | *UASp-Stg* | BDSC | 58439 | |
| Genetic reagent (*D. melanogaster*) | *UASp-tefu-RNAi* | THFC | THU5591 | |
| Genetic reagent (*D. melanogaster*) | *UASp-uev1a-RNAi* | BDSC | 66947 | |
| Genetic reagent (*D. melanogaster*) | *UASz-flag-Ras$^{G12V}$* | This paper | | Construction information described in the Materials and methods section |
| Genetic reagent (*D. melanogaster*) | *UASz-UBE2V1* | This paper | | Construction information described in the Materials and methods section |
| Genetic reagent (*D. melanogaster*) | *UASz-UBE2V2* | This paper | | Construction information described in the Materials and methods section |
| Genetic reagent (*D. melanogaster*) | *UASz-uev1a* | This paper | | Construction information described in the Materials and methods section |
| Genetic reagent (*D. melanogaster*) | *UASz-Yki$^{3SA}$* | **Zhang et al., 2024a** | | |
| Genetic reagent (*D. melanogaster*) | *nos-int; attP40* | BDSC | 79604 | |
| Genetic reagent (*D. melanogaster*) | *uev1a$^{Δ1}$* | This paper | | Construction information described in the Materials and methods section |
| Genetic reagent (*D. melanogaster*) | *uev1a$^{Δ2}$* | This paper | | Construction information described in the Materials and methods section |
| Genetic reagent (*D. melanogaster*) | *uev1a-flag* | This paper | | Construction information described in the Materials and methods section |
| Genetic reagent (*D. melanogaster*) | All deficiency *Drosophila* strains (including 7584) | The Core Facility of Drosophila Resource and Technology (CEMCS), Chinese Academy of Sciences (CAS), China | The stock numbers are the same as in BDSC | |
| Cell line (*D. melanogaster*) | Schneider 2 (S2) cells | Beyotime Biotechnology | Cat# C7925, RRID:CVCL_Z232 | |
| Cell line (*Homo sapiens*) | 293T cells | The American Type Culture Collection (ATCC) | Cat# CRL-3216, RRID:CVCL_0063 | |
| Cell line (*H. sapiens*) | HCT116 cells | ATCC | Cat# CCL-247, RRID:CVCL_VU38 | |
| Cell line (*H. sapiens*) | SW480 cells | ATCC | Cat# CCL-228, RRID:CVCL_0546 | |
| Transfected construct (*H. sapiens*) | Control shRNA | This paper | | CCTAAGGTTAAGTCGCCCTCG |
| Transfected construct (*H. sapiens*) | *UBE2V1* shRNA #1 | This paper | | CTCGGGCAGATGACATGAAAT |
| Transfected construct (*H. sapiens*) | *UBE2V1* shRNA #2 | This paper | | GCATCACCACAGGCTGGCTCA |
| Transfected construct (*H. sapiens*) | *UBE2V2* shRNA #1 | This paper | | GTCTTAAATCAACAACCTTCT |
| Transfected construct (*H. sapiens*) | *UBE2V2* shRNA #2 | This paper | | GCTCCTCCGTCAGTTAGATTT |
| Antibody | Anti-α-Spectrin (Mouse monoclonal) | Developmental Studies Hybridoma Bank (DSHB) | RRID:AB_528473 | IF (1:100) |
| Antibody | Anti-β-Actin (Mouse monoclonal) | Abmart | RRID:AB_2936240 | WB (1:5000) |

*Appendix 1 Continued*

| Reagent type (species) or resource | Designation | Source or reference | Identifiers | Additional information |
|---|---|---|---|---|
| Antibody | Anti-β-Actin (Mouse monoclonal) | Zenbio | Cat# 200068-8F10 | WB (1:5000) |
| Antibody | Anti-CycA (Rabbit polyclonal) | *Whitfield et al., 1990* | | IF (1:1000) |
| Antibody | Anti-Flag (Mouse monoclonal) | Sigma | Cat# F1804, RRID:AB_262044 | IF (1:500) |
| Antibody | Anti-Flag (Mouse monoclonal) | Utibody | Cat# UM3009 | IP (1:200), WB (1:5000) |
| Antibody | Anti-γH2AV (Mouse monoclonal) | DSHB | PRID: AB_2618077 | IF (1:200) |
| Antibody | Anti-HA (Mouse monoclonal) | Utibody | Cat# UM3004 | IP (1:200), WB (1:3000) |
| Antibody | Anti-Myc (Mouse monoclonal) | Utibody | Cat# UM3011 | IP (1:200), WB (1:3000) |
| Antibody | Anti-UBE2V2 (Mouse monoclonal) | Santa Cruz Biotechnology | Cat# sc-377254 | WB (1:2000) |
| Antibody | Anti-α-Tubulin (Rabbit polyclonal) | Proteintech | Cat# 14555-1-AP | WB (1:5000) |
| Antibody | Anti-β-Tubulin (Rabbit polyclonal) | Zenbio | Cat# 380628 | WB (1:5000) |
| Antibody | Anti-CycA (Rabbit polyclonal) | Immunoway | Cat# YT1167 | IF (1:200) |
| Antibody | Anti-Flag (Rabbit monoclonal) | Zenbio | Cat# R24091 | IP (1:200), WB (1:5000) |
| Antibody | Anti-HA (Rabbit monoclonal) | Zenbio | Cat# 301113 | IP (1:200), WB (1:3000) |
| Antibody | Anti-Ki67 (Rabbit polyclonal) | Proteintech | Cat# 27309-1-AP | IF (1:2000) |
| Antibody | Anti-Myc (Rabbit polyclonal) | ABclonal | Cat# AE009 | IP (1:200), WB (1:3000) |
| Antibody | Anti-UBE2V1 (Rabbit polyclonal) | Wanleibio | Cat# WL04482 | WB (1:2000) |
| Antibody | Alexa Fluor 546 goat anti-mouse | Invitrogen | Cat# A-11030 | IF (1:2000) |
| Antibody | HRP Goat anti-Mouse IgG(H+L) | SIMUBIOTECH | Cat# S2002 | IF (1:2000) |
| Antibody | HRP Goat anti-Rabbit IgG(H+L) | SIMUBIOTECH | Cat# S2001 | IF (1:2000) |
| Recombinant DNA reagent | *pCDH-CMV* | Addgene | RRID:Addgene_72265 | |
| Recombinant DNA reagent | *pCDH-CMV-UBE2V1* | This paper | | Construction information described in the Materials and methods section |
| Recombinant DNA reagent | *pCDH-CMV-UBE2V2* | This paper | | Construction information described in the Materials and methods section |
| Recombinant DNA reagent | *pCFD3* | Addgene | RRID:Addgene_49410 | |
| Recombinant DNA reagent | *pCFD3-uev1a-gRNA-1* | This paper | | Construction information described in the Materials and methods section |
| Recombinant DNA reagent | *pLKO-CMV-puro* | Addgene | RRID:Addgene_131700 | |

*Appendix 1 Continued on next page*

*Appendix 1 Continued*

| Reagent type (species) or resource | Designation | Source or reference | Identifiers | Additional information |
|---|---|---|---|---|
| Recombinant DNA reagent | pLKO-CMV-copGFP-puro-shNC | This paper | | Construction information described in the Materials and methods section |
| Recombinant DNA reagent | pLKO-CMV-copGFP-puro-shUBE2V1-#1 | This paper | | Construction information described in the Materials and methods section |
| Recombinant DNA reagent | pLKO-CMV-copGFP-puro-shUBE2V1-#2 | This paper | | Construction information described in the Materials and methods section |
| Recombinant DNA reagent | pLKO-CMV-copGFP-puro-shUBE2V2-#1 | This paper | | Construction information described in the Materials and methods section |
| Recombinant DNA reagent | pLKO-CMV-copGFP-puro-shUBE2V2-#2 | This paper | | Construction information described in the Materials and methods section |
| Recombinant DNA reagent | pMD2.G | Addgene | RRID:Addgene_12259 | |
| Recombinant DNA reagent | psPAX2 | Addgene | RRID:Addgene_12260 | |
| Recombinant DNA reagent | pU6-BbsI-chiRNA | Addgene | RRID:Addgene_45946 | |
| Recombinant DNA reagent | pU6-uev1a-gRNA-2 | This paper | | Construction information described in the Materials and methods section |
| Recombinant DNA reagent | pUASt-attB | Drosophila Genomics Resource Center (DGRC) | RRID:DGRC_1419 | |
| Recombinant DNA reagent | pUASt-APC7-Myc | This paper | | Construction information described in the Materials and methods section |
| Recombinant DNA reagent | pUASt-Flag-ben | This paper | | Construction information described in the Materials and methods section |
| Recombinant DNA reagent | pUASt-Flag-cycA | This paper | | Construction information described in the Materials and methods section |
| Recombinant DNA reagent | pUASt-HA-Ub | This paper | | Construction information described in the Materials and methods section |
| Recombinant DNA reagent | pUASt-HA-Ub$^{7KR}$ | This paper | | Construction information described in the Materials and methods section |
| Recombinant DNA reagent | pUASt-HA-Ub$^{6KR+K11}$ | This paper | | Construction information described in the Materials and methods section |
| Recombinant DNA reagent | pUASt-HA-Ub$^{6KR+K48}$ | This paper | | Construction information described in the Materials and methods section |
| Recombinant DNA reagent | pUASt-HA-Ub$^{6KR+K63}$ | This paper | | Construction information described in the Materials and methods section |
| Recombinant DNA reagent | pUASt-HA-uev1a | This paper | | Construction information described in the Materials and methods section |
| Recombinant DNA reagent | pUASt-Myc-APC4 | This paper | | Construction information described in the Materials and methods section |
| Recombinant DNA reagent | pUASt-Myc-Cdc16 | This paper | | Construction information described in the Materials and methods section |
| Recombinant DNA reagent | pUASt-Myc-Cdc23 | This paper | | Construction information described in the Materials and methods section |
| Recombinant DNA reagent | pUASt-Myc-Cdc27 | This paper | | Construction information described in the Materials and methods section |
| Recombinant DNA reagent | pUASt-Myc-Fzr | This paper | | Construction information described in the Materials and methods section |
| Recombinant DNA reagent | pUASt-Myc-Fzy | This paper | | Construction information described in the Materials and methods section |

*Appendix 1 Continued on next page*

*Appendix 1 Continued*

| Reagent type (species) or resource | Designation | Source or reference | Identifiers | Additional information |
|---|---|---|---|---|
| Recombinant DNA reagent | pUASt-Myc-Ida | This paper | | Construction information described in the Materials and methods section |
| Recombinant DNA reagent | pUASt-Myc-Mr | This paper | | Construction information described in the Materials and methods section |
| Recombinant DNA reagent | pUASz1.0 | DGRC | RRID:DGRC_1431 | |
| Recombinant DNA reagent | pUASz-flag-Ras$^{G12V}$ | This paper | | Construction information described in the Materials and methods section |
| Recombinant DNA reagent | pUASz-UBE2V1 | This paper | | Construction information described in the Materials and methods section |
| Recombinant DNA reagent | pUASz-UBE2V2 | This paper | | Construction information described in the Materials and methods section |
| Recombinant DNA reagent | pUASz-uev1a | This paper | | Construction information described in the Materials and methods section |
| Recombinant DNA reagent | pUASz-uev1a-donor | This paper | | Construction information described in the Materials and methods section |
| Recombinant DNA reagent | pUASz-Yki$^{3SA}$ | This paper | | Construction information described in the Materials and methods section |
| Sequence-based reagent | uev1a#1_F | This paper | PCR primers | GAATTAATACGACTCACTATAGG GAGAACGGAATTTCCGCTTACTG |
| Sequence-based reagent | uev1a#1_R | This paper | PCR primers | GAATTAATACGACTCACTATAGG GAGACGGACCGATGATCATGCC |
| Sequence-based reagent | uev1a#2_F | This paper | PCR primers | GAATTAATACGACTCACTATAGGG AGACACTAAAGATCGAGTGCG |
| Sequence-based reagent | uev1a#2_R | This paper | PCR primers | GAATTAATACGACTCACTATAGG GAGATGCCAGCTTCAGGTTCTC |
| Sequence-based reagent | ben#1_F | This paper | PCR primers | GAATTAATACGACTCACTATAGG GAGACCACGTCGCATCATCAAG |
| Sequence-based reagent | ben#1_R | This paper | PCR primers | GAATTAATACGACTCACTATAGGG AGAAGTCTTCGACGGCATATTTC |
| Sequence-based reagent | ben#2_F | This paper | PCR primers | GAATTAATACGACTCACTATAGG GAGACAGATCCGGACCATATTG |
| Sequence-based reagent | ben#2_R | This paper | PCR primers | GAATTAATACGACTCACTATAGGG AGATCAGTCTTCGACGGCATATTTC |
| Sequence-based reagent | cdc27#1_F | This paper | PCR primers | GAATTAATACGACTCACTATAGG GAGATCGCCCAGGATCTGATTAAC |
| Sequence-based reagent | cdc27#1_R | This paper | PCR primers | GAATTAATACGACTCACTATAGGG AGAGCAGCGACAGATCCTTCTTC |
| Sequence-based reagent | cdc27#2_F | This paper | PCR primers | GAATTAATACGACTCACTATAGGG AGAGATGATGGGCAAAAAGCTAAAG |
| Sequence-based reagent | cdc27#2_R | This paper | PCR primers | GAATTAATACGACTCACTATAGG GAGACCATCGGCCGATTGTTTC |
| Sequence-based reagent | AcGFP-F | This paper | PCR primers | GAATTAATACGACTCACTATAGGGAG ATGCACCACCGGCAAGCTGCCTG |
| Sequence-based reagent | AcGFP-R | This paper | PCR primers | GAATTAATACGACTCACTATAGGGA GAGGCCAGCTGCACGCTGCCATC |
| Sequence-based reagent | GAPDH-F | This paper | PCR primers | ACAACTTTGGTATCGTGGAAGG |
| Sequence-based reagent | GAPDH-R | This paper | PCR primers | GCCATCACGCCACAGTTTC |
| Sequence-based reagent | UBE2V1-F | This paper | PCR primers | CGGGCTCGGGAGTAAAAGTC |
| Sequence-based reagent | UBE2V1-R | This paper | PCR primers | AGGCCCAATTATCATCCCTGT |

*Appendix 1 Continued on next page*

*Appendix 1 Continued*

| Reagent type (species) or resource | Designation | Source or reference | Identifiers | Additional information |
|---|---|---|---|---|
| Sequence-based reagent | *UBE2V2*-F | This paper | PCR primers | TGGACAGGCATGATTATTGGGC |
| Sequence-based reagent | *UBE2V2*-R | This paper | PCR primers | CTAACACTGGTATGCTCCGGG |
| Commercial assay or kit | BCA Protein Assay Kit | Beyotime Biotechnology | Cat# P0012 | |
| Commercial assay or kit | CCK8 Kit | APExBIO | Cat# K1018 | |
| Commercial assay or kit | EdU Incorporation Assay Kit | Beyotime Biotechnology | Cat# C0075 | |
| Commercial assay or kit | SPARKscript II All-in-one RT SuperMix kit | SparkJade | Cat# AG0305-C | |
| Commercial assay or kit | SYBR Green Premix Pro Taq HS qPCR kit | ACCURATE BIOLOGY | Cat# AG11701-S | |
| Commercial assay or kit | T7 RiboMAX Express RNAi System | Promega | Cat# P1700 | |
| Commercial assay or kit | TriQuick Reagent kit | Solarbio | Cat# R1100 | |
| Chemical compound, drug | Chloroquine (CQ) | Selleck | Cat# S6999 | |
| Chemical compound, drug | Cycloheximide (CHX) | MCE | Cat# HY-12320 | |
| Chemical compound, drug | MG132 | Selleck | Cat# S2619 | |
| Chemical compound, drug | Protein G Sepharose | Cytiva | Cat# 17061801 | |
| Chemical compound, drug | Puromycin | Solarbio | Cat# P8230 | |
| Software, algorithm | Adobe Photoshop 2022 | San Jose, CA, USA | RRID:SCR_014199 | |
| Software, algorithm | ImageJ | NIH | RRID:SCR_003070 | |
| Software, algorithm | GraphPad Prism | GraphPad Software, Inc | RRID:SCR_002798 | |
| Other | DMSO | Macklin | Cat# D6258 | |
| Other | Dulbecco's Modified Eagle Medium | Gibco | Cat# C11995500BT | |
| Other | Fetal Bovine Serum (FBS) | Lonsera | Cat# S711-001S | |
| Other | Insect Culture Medium | Union | Cat# UK1000 | |
| Other | Lipofectamine 2000 | Thermo Fisher | 11668027 | |

