## [Editor Report · eLife Assessment]

This **valuable** study examines the role of E2 ubiquitin enzyme, Uev1a in tissue resistance to oncogenic RasV12 in *Drosophila melanogaster* polyploid germline cells and human cancer cell lines. The **solid** evidence suggests that Uev1a works with the E3 ligase APC/C to degrade Cyclin A. This work would be of interest to researchers in germline biology and cancer.

---

## [Referee Report · Reviewer #1 (Public review)]

Summary:

This study uncovers a protective role of the ubiquitin-conjugating enzyme variant Uev1A in mitigating cell death caused by over-expressed oncogenic Ras in polyploid *Drosophila* nurse cells and by RasK12 in diploid human tumor cell lines. The authors previously showed that over-expression of oncogenic Ras induces death in nurse cells, and now they perform a deficiency- screen for modifiers. They identified Uev1A as a suppressor of this Ras-induced cell death. Using genetics and biochemistry, the authors found that Uev1A collaborates with the APC/C E3 ubiquitin ligase complex to promote proteasomal degradation of Cyclin A. This function of Uev1A appears to extend to diploid cells, where its human homologs UBE2V1 and UBE2V2 suppress oncogenic Ras-dependent phenotypes in human colorectal cancer cells in vitro and in xenografts in mice.

Strengths:

(1) Most of the data is supported by sufficient sample size and appropriate statistics.

(2) Good mix of genetics and biochemistry.

(3) Generation of new transgenes and *Drosophila* alleles that will be beneficial for the community.

Comments on revisions:

The authors have greatly improved the manuscript and satisfactorily addressed all of my concerns.

---

## [Referee Report · Reviewer #2 (Public review)]

Summary:

The authors performed a genetic screen using deficiency lines and identified Uev1a as a factor that protects nurse cells from RasG12V-induced cell death. According to a previous study from the same lab, this cell death is caused by aberrant mitotic stress due to CycA upregulation (Zhang et al.). This paper further reveals that Uev1a forms a complex with APC/C to promote proteasome-mediated degradation of CycA.

In addition to polyploid nurse cells, the authors also examined the effect of RasG12V-overexpression in diploid germline cells, where RasG12V-overexpression triggers active proliferation not cell death. Uev1a was found to suppress its overgrowth as well.

Finally, the authors show that the overexpression of the human homolog, UBE2V1 and UBE2V2, suppresses tumor growth in human colorectal cancer xenografts and cell lines. Notably, these genes' expression correlates with the survival of colorectal cancer patients carrying Ras mutation.

Strength:

This paper presents a significant finding that UBE2V1/2 may serve as a potential therapy for cancers harboring Ras mutations. The authors propose a fascinating mechanism in which Uev1a forms a complex with APC/C to inhibit aberrant cell cycle progression.

Comments on revisions:

The authors have addressed several of the major concerns, including the addition of new data and improved figure presentation. However, some issues remain insufficiently resolved, particularly regarding control reuse (Major Comment 3) and experimental interpretation (Major Comments 5 and 8).

Regarding Major Comment 5, the authors state that UAS copy number affects the frequency of egg chamber degradation in Fig. 2D, and thus explains the reduced phenotype in RasG12V + GFP-RNAi compared to RasG12V alone. However, this explanation is not consistent with other data in the manuscript. UAS-RasG12V combined with UAS-lacZ in Fig. 2G shows a phenotype comparable to UAS-RasV12 alone, despite also increasing the UAS copy number. This suggests that the effect is not simply due to copy number.

I understand that the authors used UAS-RasG12V + GFP-RNAi as a control for the RNAi experiments and UAS-RasG12V + lacZ for the overexpression experiments. I suggest examining the phenotype frequency of UAS-RasG12V + UAS-GFP, to figure the reason out. Overall, these results indicate that there is a spectrum of phenotype frequencies, and therefore appropriate controls should be included for each experiment rather than reusing the same dataset across different experiments, as also noted in Major Comment 3.

---

## [Author Response]

The following is the authors’ response to the original reviews.

**eLife Assessment**
This valuable study examines the role of E2 ubiquitin enzyme, Uev1a in tissue resistance to oncogenic RasV12 in *Drosophila melanogaster* polyploid germline cells and human cancer cell lines. The incomplete evidence suggests that Uev1a works with the E3 ligase APC/C to degrade Cyclin A, and the strength of evidence could be increased by addressing the expression of CycA in the ovaries and the uev1a loss of function in human cancer cells. This work would be of interest to researchers in germline biology and cancer.

Thank you for your valuable assessment. The requested data on CycA expression (Figure 4E-G) and *uev1a* loss-of-function in human cancer cells (Figure 8 and Figure 8-figure supplement 2) have been added to the revised manuscript.

**Public Reviews:**

**Reviewer #1 (Public review):**
Summary:This study uncovers a protective role of the ubiquitin-conjugating enzyme variant Uev1A in mitigating cell death caused by over-expressed oncogenic Ras in polyploid *Drosophila* nurse cells and by RasK12 in diploid human tumor cell lines. The authors previously showed that overexpression of oncogenic Ras induces death in nurse cells, and now they perform a deficiency screen for modifiers. They identified Uev1A as a suppressor of this Ras-induced cell death. Using genetics and biochemistry, the authors found that Uev1A collaborates with the APC/C E3 ubiquitin ligase complex to promote proteasomal degradation of Cyclin A. This function of Uev1A appears to extend to diploid cells, where its human homologs UBE2V1 and UBE2V2 suppress oncogenic Ras-dependent phenotypes in human colorectal cancer cells in vitro and in xenografts in mice.Strengths:(1) Most of the data is supported by a sufficient sample size and appropriate statistics.(2) Good mix of genetics and biochemistry.(3) Generation of new transgenes and *Drosophila* alleles that will be beneficial for the community.

We greatly appreciate your comments.

Weaknesses:(1) Phenotypes are based on artificial overexpression. It is not clear whether these results are relevant to normal physiology.

Downregulation of Uev1A, Ben, and Cdc27 together significantly increased the incidence of dying nurse cells in normal ovaries (Figure 5-figure supplement 2), indicating that the mechanism we uncovered also protects nurse cells from death during normal oogenesis.

(2) The phenotype of "degenerating ovaries" is very broad, and the study is not focused on phenotypes at the cellular level. Furthermore, no information is provided in the Materials and Methods on how degenerating ovaries are scored, despite this being the most important assay in the study.

Thank you for pointing out this issue. We quantified the phenotype of nurse cell death using “degrading/total egg chambers per ovary”, not “degenerating ovaries”. Normal nurse cell nuclei exhibit a large, round morphology in DAPI staining (see the first panel in Figure 1D). During early death, they become disorganized and begin to condense and fragment (see the second panel in Figure 1D). In late-stage death, they are completely fragmented into small, spherical structures (see the third panel in Figure 1D), making cellular-level phenotypic quantification impossible. Since all nurse cells within the same egg chamber are interconnected, their death process is synchronous. Thus, quantifying the phenotype at the egg-chamber level is more practical than at the cellular level. We have added the description of this death phenotype and its quantification to the main text (Lines 104-108).

(3) In Figure 5, the authors want to conclude that uev1a is a tumor-suppressor, and so they over-express ubev1/2 in human cancer cell lines that have RasK12 and find reduced proliferation, colony formation, and xenograft size. However, genes that act as tumor suppressors have loss-of-function phenotypes that allow for increased cell division. The *Drosophila* uev1a mutant is viable and fertile, suggesting that it is not a tumor suppressor in flies. Additionally, they do not deplete human ubev1/2 from human cancer cell lines and assess whether this increases cell division, colony formation, and xenograph growth.

We apologize for any misleading description. We aimed to demonstrate that UBE2V1/2, like Uev1A in *Drosophila* “*nos>RasG12V+bam-RNAi*” germline tumors, suppress oncogenic *KRAS*-driven overgrowth in diploid human cancer cells. Importantly, this function of Uev1A and UBE2V1/2 is dependent on *Ras*-driven tumors; there is no evidence that they act as broad tumor suppressors in the absence of oncogenic *Ras*. *Drosophila uev1a* mutants were lethal, not viable (see Lines 135-137), and germline-specific knockdown of *uev1a (nos>uev1a-RNAi)* caused female sterility without inducing tumors. These findings suggest that Uev1A lacks tumor-suppressive activity in the *Drosophila* female germline in the absence of *Ras*-driven tumors. We have revised the manuscript to prevent misinterpretation. Furthermore, we have added data demonstrating that the combined knockdown of *UBE2V1* and *UBE2V2* significantly promotes the growth of *KRAS*-mutant human cancer cells, as suggested (Figure 8 and Figure 8-figure supplement 2).

(4) A critical part of the model does not make sense. CycA is a key part of their model, but they do not show CycA protein expression in WT egg chambers or in their over-expression models (nos.RasV12 or bam>RasV12). Based on Lilly and Spradling 1996, Cyclin A is not expressed in germ cells in region 2-3 of the germarium; whether CycA is expressed in nurse cells in later egg chambers is not shown but is critical to document comprehensively.

We appreciate your critical comment. CycA is a key cyclin that partners with Cdk1 to promote cell division (Edgar and Lehner, 1996). Notably, nurse cells are post-mitotic endocycling cells (Hammond and Laird, 1985) and typically do not express CycA (Lilly and Spradling, 1996) (see the last sentence, page 2518, paragraph 3 in this 1996 paper). However, their death induced by oncogenic *RasG12V* is significantly suppressed by monoallelic deletion of either *cycA* or cdk1 (Zhang et al., 2024). Conversely, ectopic CycA expression in nurse cells triggers their death (Figure 4C, D). These findings suggest that polyploid nurse cells exhibit high sensitivity to aberrant division-promoting stress, which may represent a distinct form of cellular stress unique to polyploid cells. In the revised manuscript, we have provided the CycA-staining data, comparing its expression in normal nurse cells versus cells undergoing oncogenic *RasG12V*-induced death (Figure 4E-G).

(5) The authors should provide more information about the knowledge base of uev1a and its homologs in the introduction.

Thank you for your suggestion. In the revised introduction, we have provided a more detailed description of Uev1A (Lines 72-79). Additionally, we have introduced its human homologs, UBE2V1 and UBE2V2, in the main text (Lines 143-145).

**Reviewer #2 (Public review):**
Summary:The authors performed a genetic screen using deficiency lines and identified Uev1a as a factor that protects nurse cells from RasG12V-induced cell death. According to a previous study from the same lab, this cell death is caused by aberrant mitotic stress due to CycA upregulation (Zhang et al.). This paper further reveals that Uev1a forms a complex with APC/C to promote proteasome-mediated degradation of CycA.In addition to polyploid nurse cells, the authors also examined the effect of RasG12V-overexpression in diploid germline cells, where RasG12V-overexpression triggers active proliferation, not cell death. Uev1a was found to suppress its overgrowth as well.Finally, the authors show that the overexpression of the human homologs, UBE2V1 and UBE2V2, suppresses tumor growth in human colorectal cancer xenografts and cell lines. Notably, the expression of these genes correlates with the survival of colorectal cancer patients carrying the Ras mutation.Strength:This paper presents a significant finding that UBE2V1/2 may serve as a potential therapy for cancers harboring Ras mutations. The authors propose a fascinating mechanism in which Uev1a forms a complex with APC/C to inhibit aberrant cell cycle progression.

We greatly appreciate your comments.

Weakness:The quantification of some crucial experiments lacks sufficient clarity.

Thank you for highlighting this issue. We have provided more details regarding the quantification data in the revised manuscript.

References

Edgar, B.A., and Lehner, C.F. (1996). Developmental control of cell cycle regulators: a fly's perspective. Science 274, 1646-1652.

Hammond, M.P., and Laird, C.D. (1985). Chromosome structure and DNA replication in nurse and follicle cells of *Drosophila melanogaster*. Chromosoma 91, 267-278.

Lilly, M.A., and Spradling, A.C. (1996). The *Drosophila* endocycle is controlled by Cyclin E and lacks a checkpoint ensuring S-phase completion. Genes Dev 10, 2514-2526.

Zhang, Q., Wang, Y., Bu, Z., Zhang, Y., Zhang, Q., Li, L., Yan, L., Wang, Y., and Zhao, S. (2024). Ras promotes germline stem cell division in *Drosophila* ovaries. Stem Cell Reports 19, 1205-1216.

**Recommendations for the authors:**

**Reviewer #1 (Recommendations for the authors):**
(1) The figure legends insufficiently describe the figures. One example is Figure 3, where there are no details in the figure legend about what conditions apply to each panel and each lane of the gels.

For clarity and brevity, detailed experimental conditions are described in the Materials and Methods section. Figure legends therefore focus on summarizing the key findings. Thank you for your understanding!

(2) The font size on the figure is too small.

Thank you for your constructive suggestion. In response, we have enlarged all font sizes to improve readability.

(3) There are places where the authors overstate their results, and there are issues with the clarity of the text:(3a) Lines 170: "excessive" is not appropriate. Their prior study showed a mild increase in proliferation.

“Excessive” has been removed in the revised manuscript (Lines 215-216).

(3b) Line 187-8: The authors should restate this sentence. Here's a possibility. Over-expression of Uev1a suppressed the phenotypes caused by CycA over-expression.

This sentence has been restated as “Notably, this cell death was suppressed by co-overexpression of CycA and Uev1A, indicating a genetic interaction between them”. (Lines 229-231).

(3c) Lines 266-7: The properties of Uev1a (ie, lacking a conserved Cys) should be in the introduction.

This information has been added to the revised introduction (Lines 74-76).

(3d) Line 318: "markedly" is an overstatement of the prior results.

Our quantification data revealed that “*nos>RasG12V; bam-/-*” ovaries are three times larger than “*nos>GFP; bam-/-*” control ovaries (see Figure 4A-C in Zhang et al., Stem Cell Reports 19, 1205-1216). Given this substantial difference, we think that using "markedly" is not an overstatement.

(4) Data not shown occurs in a few places in the text. Given the ability to supply supplemental information in eLife preprints, these data should be shown.

Thanks for your suggestion. All “not shown” data have been added to the revised manuscript.

**Reviewer #2 (Recommendations for the authors):**
Major Comments(1) Cyclin A (CycA) is a key player in this study, but the authors do not provide evidence showing the upregulation of CycA following Ras overexpression in either polyploid or diploid cells. Data on CycA expression should be included.

Thank you for your constructive suggestion. These data have been added to the revised manuscript (Figure 4E-G).

(2) DNA replication stress, cellular senescence, and cell death should be assessed under Ras overexpression (RasOE) and RasOE + Uev1A RNAi conditions to support the model proposed in Figure 4F.

We apologize for any confusion caused by our initial model. We do not have evidence that DNA replication stress and cellular senescence occur under these conditions. Cell death can be readily detected through the presence of fragmented nuclei and condensed DNA (see Figure 1D). The model has been updated accordingly (Figure 9E).

(3) Appropriate controls should be performed alongside the experimental sets. The same nos>Ras+GFPi data set was repeatedly used in Figures 1I, 2B, 2H, and Figures 2, S2B, which is not ideal.

All these experiments were performed under identical conditions. Therefore, we deem it appropriate to use the same control data across these analyses.

(4) Overall, the microscopic images are too small and hard to see.

Thank you for raising this important point. In the revised manuscript, all images and the font size on figures have been enlarged for improved clarity.

(5) Figure 1HWhy is the frequency of egg chamber degradation quite less in nos>RasG12V+GFP-RNAi (about 40%) than nos > RasG12V (about 80%)? And the authors do not show that there is a significant difference between those two conditions, although it should be there. We will need the explanation from the authors on why there is a difference here.

These overexpression experiments were conducted using the *GAL4/UAS* system. While both “*nos>RasG12V+GFP-RNAi*” and “*nos>RasG12V*” contain a single *nos-GAL4* driver, they differ in *UAS* copy number: the former incorporates two *UAS* elements compared to only one in the latter (see the detailed genotypes in Source data 2). These results demonstrate that *UAS* copy number impacts experimental outcomes in our system.

In the previous paper Zhang et al. (2024), Figure 7H shows that the frequency of egg chambers in nos>RasG12V is 33%, although this paper shows it as about 80%. There seems to be a difference in flies' age (previous paper: 7d, this paper: 3d), but this data raises the question of why nos>RasG12V shows more egg chamber degradation this time.

We greatly appreciate your careful observation. The nurse-cell-death phenotype exhibits a spectrum from mild to severe manifestations [see Figure 1D and our response to weekness (2) in Reviewer #1’s public reviews]. While our 2024 paper exclusively quantified egg chambers with severe phenotypes as degrading, the current study included both mild and severe cases in this classification. We do not think fly age could account for this substantial phenotypic difference. A detailed description of the nurse-cell-death phenotype and its quantification have been added to the revised manuscript (Lines 104-108).

In the following experiments, only nos>RasG12V+GFP-RNAi is used as a control (Figures 2B, H, S2B). I wonder if these results would give us a different conclusion if nos>RasG12V were used as a control.

As explained above, the *UAS* copy number does matter in our analyses, so it is important to keep them identical for comparison.

(6) In the abstract, the authors mention that uev1a is an intrinsic factor to protect cells from RasG12V-induced cell death. RasG12V does not induce much cell death of cystocytes with bam-gal4, whereas it induces a lot of nurse cells' death. Does it mean the intrinsic expression level of uev1a is low in nurse cells (or polyploid cells) compared to cystocytes (or diploid cells)?

Overexpression of Ras^G12V^ driven by *bam-GAL4* exhibited only minimal nurse cell death (Figure 1D, E). Additionally, Uev1A exhibited low intrinsic expression levels in both cystocytes and nurse cells (Figure 3E and Figure 5-figure supplement 1).

(7) Is uev1a-RNAi alone sufficient to induce egg chamber degradation? Or does it have any effect on ovarian development? (Related to question #1 in minor comments)

While *nos>uev1a-RNAi* resulted in female sterility, it alone was insufficient to induce egg chamber degradation. However, simultaneous downregulation of Uev1A, Ben, and Cdc27 triggered significant egg chamber degradation (Figure 5-figure supplement 2).

(8) Which stages of egg chambers get degraded with RasG12V induction?

This is a good question. In our analyses, we noted that degrading egg chambers exhibited considerable size variability (Figure 1D). Because degradation disrupts normal morphological cues, precise staging of these egg chambers is nearly impossible.

(9) I suggest testing the cellular senescence marker as well if the authors mention that CycA-degradation by Uev1a-APC/C complex prevents cellular senescence induced by RasG12V in a schematic image of Figure 4 (e.g., Dap/p21, SA-β-gal).

As addressed in our response to your Major Comment (2), we lacked experimental evidence to support cellular senescence in this context. We have therefore revised the model accordingly (Figure 9E). While this study focuses specifically on cell death, investigating potential roles of cellular senescence remains an important direction for future research. Thank you for your suggestion!

Minor Comments(1) Figure 1D: Df#7584It seems that the late-stage egg chamber is missing in this condition. Why does this occur without egg chamber degradation? Is there a possibility that we do not see egg chamber degradation because this deficiency line does not have a properly developed egg chamber that can have a degradation?

While this image represents only a single sample, we have confirmed the presence of late-stage egg chambers in other samples. If “*Df#7584/+*” females were unable to support late-stage egg chamber development, complete sterility would be expected due to the lack of mature eggs. However, as shown in this image (Figure 1D), the ovary contains mature eggs, and the “*Df#7584/+*” fly strain remains fertile.

(2) Based on the results that DDR signaling functions as keeping egg chambers from degradation, the authors may be better to check the DNA-damage markers in nos>RasG12V, nos>RasG12V +uev1a. (e.g. γ-H2AX)

Thank you for your constructive recommendation. These data have been added to the revised manuscript (Figure 3C).